# Structural models of genome-wide covariance identify multiple common dimensions in autism

Lucía de Hoyos [1], Maria T. Barendse [1,2], Fenja Schlag[1],
Marjolein M. J. van Donkelaar[1], Ellen Verhoef [1], Chin Yang Shapland [3,4],
Alexander Klassmann [5], Jan Buitelaar [6,7,8], Brad Verhulst [9],
Simon E. Fisher [1,6], Dheeraj Rai[4,10,11] & Beate St Pourcain [1,3,6] ✉

Common genetic variation has been associated with multiple phenotypic features in Autism Spectrum Disorder (ASD). However, our knowledge of shared genetic factor structures contributing to this highly heterogeneous phenotypic spectrum is limited. Here, we developed and implemented a structural equation modelling framework to directly model genomic covariance across core and non-core ASD phenotypes, studying autistic individuals of European descent with a case-only design. We identified three independent genetic factors most strongly linked to language performance, behaviour and developmental motor delay, respectively, studying an autism community sample (N = 5331). The three-factorial structure was largely confirmed in independent ASD-simplex families (N = 1946), although we uncovered, in addition, simplex-specific genetic overlap between behaviour and language phenotypes. Multivariate models across cohorts revealed novel associations, including links between language and early mastering of self-feeding. Thus, the common genetic architecture in ASD is multi-dimensional with overarching genetic factors contributing, in combination, with ascertainment-specific patterns, to phenotypic heterogeneity.

Autism spectrum disorder (ASD) is a complex neurodevelopmental condition with considerable phenotypic and genetic heterogeneity[1,2]. Core phenotypes in ASD implicate difficulties in social interaction and communication, as well as restricted, repetitive behavioural patterns and sensory abnormalities[3]. However, the phenotypic presentation is broad and variable (phenotypic heterogeneity). More than 70% of individuals with ASD are diagnosed with co-occurring conditions[4], and individuals differ in phenotypic presentation, especially cognitive functioning[2,4]. At the genetic level, additive genetic effects of rare and common genetic factors contribute to ASD liability[1,5–10] (genetic heterogeneity). Common variation explains most genetic variance in ASD, accounting for 12 to 65% of liability[1,5,11]. However, even common genetic variation is highly heterogenous in ASD[5,6,8], and differences in underlying shared genetic factors are only partially understood.

[1]Language and Genetics Department, Max Planck Institute for Psycholinguistics, Nijmegen, The Netherlands. [2]Department of Social Dentistry and Behavioural Sciences, Academic Centre for Dentistry Amsterdam (ACTA), Amsterdam, The Netherlands. [3]MRC Integrative Epidemiology Unit, University of Bristol, Bristol, UK. [4]Population Health Sciences, University of Bristol, Bristol, UK. [5]Institute for Genetics, University of Cologne, Cologne, Germany. [6]Donders Institute for Brain, Cognition and Behaviour, Radboud University, Nijmegen, The Netherlands. [7]Karakter Child and Adolescent Psychiatry University Centre, Nijmegen, The Netherlands. [8]Department of Cognitive Neuroscience, Radboud University Medical Center, Nijmegen, The Netherlands. [9]Texas A&M University, College Station, TX, USA. [10]Avon and Wiltshire Partnership NHS Mental Health Trust, Bristol, UK. [11]NIHR Biomedical Research Centre, University of Bristol, Bristol, UK. ✉e-mail: Beate.StPourcain@mpi.nl

Depending on an individual's genetic architecture, common variants act through partially distinct aetiological mechanisms[6]. For example, autistic individuals with intellectual disability (ID), compared to those without, carry a higher rate of contributing de novo variants[6] and show qualitative differences in their common genetic architecture[5]. In addition, polygenic scores (PGS) for different disorders, aggregating common risk alleles, show distinct association profiles with phenotypic factor structures in groups comprising only autistic individuals[8,12]. Thus, also common variation may present genetic factor structures linking phenotypic domains, although the number of factors and their nature is unknown.

In this study, we aim to understand whether phenotypic heterogeneity in ASD can be explained by heterogeneity in common genetic effects by studying autistic individuals from large ASD cohorts. To do so, we fully dissect the single nucleotide polymorphism heritability ($h^2_{SNP}$) of ASD phenotypes into shared and specific genomic variance contributions, as implemented in genetic-relationship-matrix (GRM) structural equation modelling (GRM-SEM)[13,14]. GRM-SEM, a genetic confirmatory factor analysis technique, leverages the genetic relatedness between individuals, as measured by direct genome-wide genotyping data, to model genetic and residual factor structures[13,14] in a multivariate setting. Therefore, GRM-SEM allows the direct modelling of the genomic covariance, in contrast to previous studies[8,9] that interrogate the genetic architecture in ASD through analyses of phenotypic factor structures followed by genetic association analyses. By examining core and non-core ASD phenotypes, we estimate the number of shared genetic dimensions and elucidate their underlying structure with a novel data-driven genomic covariance modelling approach, building on previous GRM-SEM efforts[13,14]. Using a case-only design, we investigate 5331 autistic individuals from the Simons

Foundation Powering Autism Research for Knowledge (SPARK) sample[15], as part of discovery analyses. We conceptually replicate our results on 1,946 autistic individuals from the Simon Simplex Collection (SSC)[16]. Our study provides new insights into the multi-dimensional common genetic architecture of ASD and shows that phenotypic heterogeneity can, largely, be captured by shared genetic factors.

## Results

### Multi-dimensional genetic analyses in community-based ASD

Genetic heterogeneity, as explained by common variation, and phenotypic heterogeneity will be most prominent in ASD community samples, i.e. unselected ASD samples with a wide demographic, phenotypic and clinical spectrum. Here, we conducted discovery analyses in SPARK[15], a cohort that represents autistic individuals from the United States from multiplex and simplex families with rich phenotypic information.

To identify shared genetic factors, we implemented a multi-stage approach (Fig. 1a, Methods). During the first stage (Stage I, Fig. 1a), we identified phenotypes that likely have genetic contributions ($h^2_{SNP}$, $p \leq 0.1$) using Genomic Restricted Maximum Likelihood (GREML)[17,18], screening a wide range of language, cognitive, motor, developmental, affective, behavioural and social phenotypes (https://www.sfari.org/spark-demographic-and-clinical-information). This increases power and ensures model convergence, as not all phenotypes may have common genetic contributions. From an initial set of 47 phenotypes (Methods, Supplementary Data 1, Supplementary Fig. 1), we retained 17 phenotypes representing five ASD domains: language/cognition, general behaviour, developmental milestones, motor, and repetitive behavioural features (Fig. 2a). Notably, social and affective phenotypes showed little $h^2_{SNP}$ (Supplementary Fig. 2), consistent with either

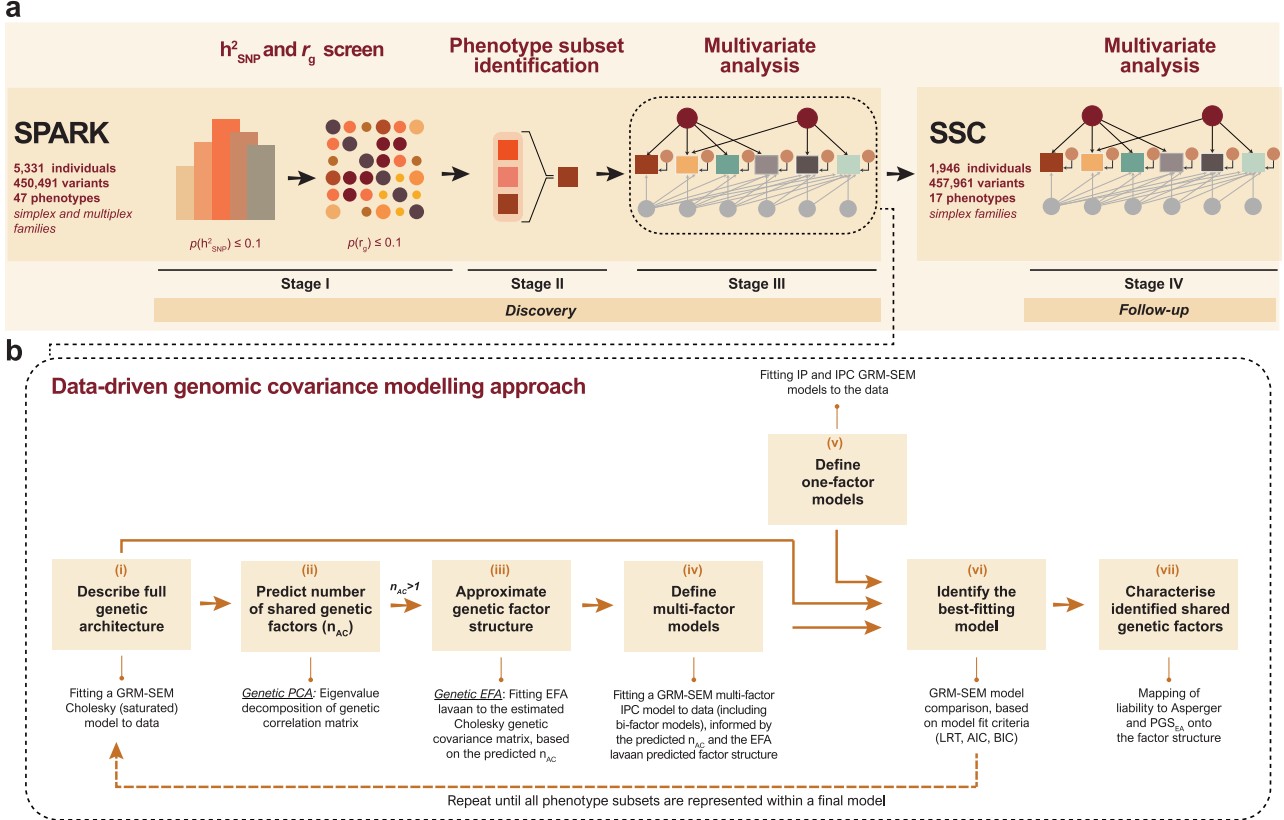

**Fig. 1 | Workflow of the study. a** Multi-stage study design. Multivariate discovery analyses were carried out in the Simons Powering Autism Research (SPARK) sample (Stages I-III) and the best-fitting model in SPARK was followed-up in the Simons Simplex Collection (SSC, Stage IV). **b** Data-driven genomic covariance modelling approach, including a step-wise combination of principal component analysis (PCA), exploratory factor analysis (EFA) and Genetic-relationship-matrix structural equation modelling (GRM-SEM), as described in the Methods.

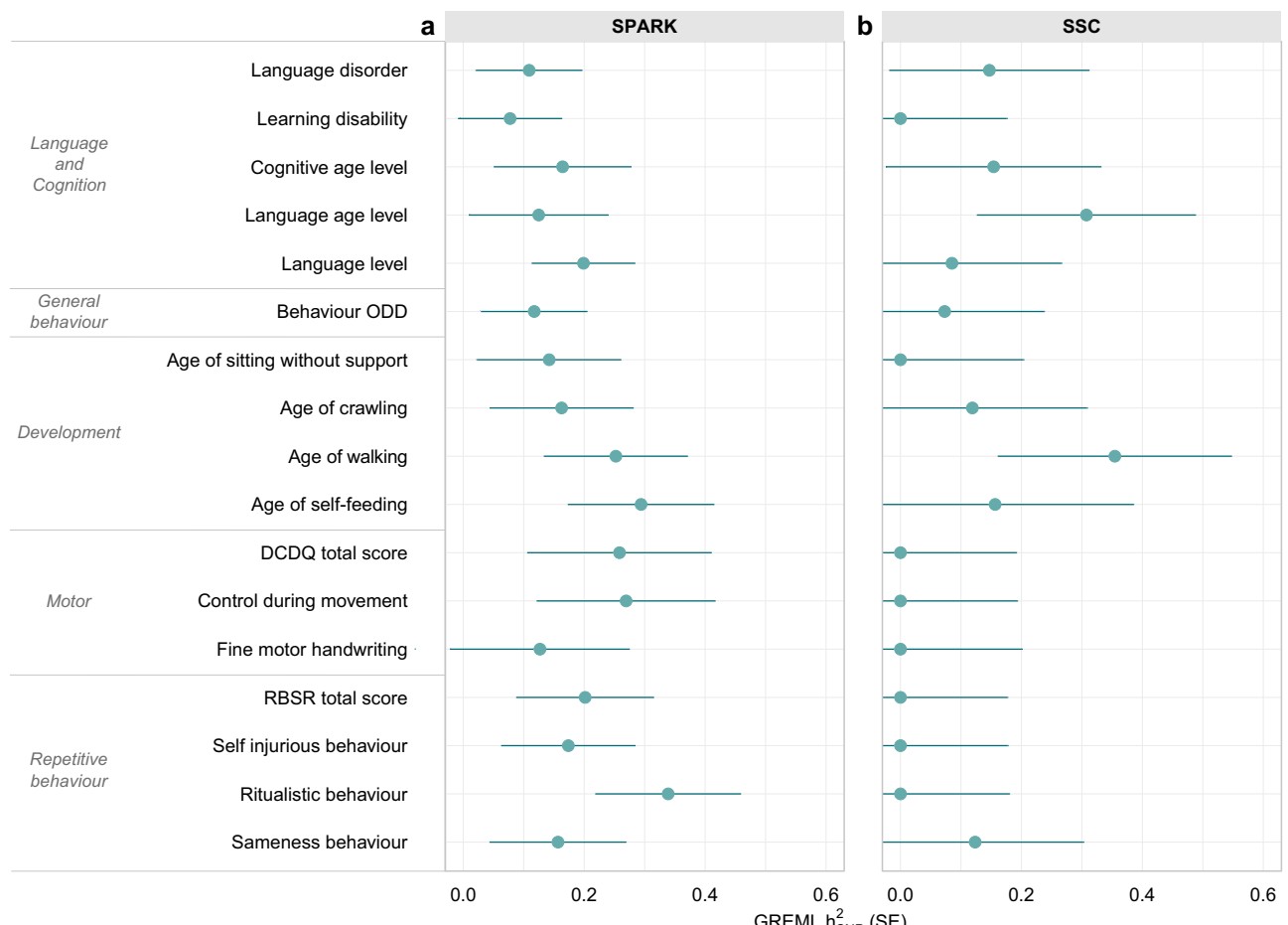

**Fig. 2 | GREML heritability estimates for SPARK and SSC phenotypes. a** GREML $h^2_{SNP}$ of continuous and categorical ASD phenotypes with $p \leq 0.1$ in the SPARK sample (N ≤ 5132). A complete figure of all analysed phenotypes is shown in Supplementary Fig. 2. Information on phenotype description, sample size and exact heritability and $p$-values is available in Supplementary Data 1. **b** GREML $h^2_{SNP}$ of continuous and categorical ASD phenotypes in the SSC sample (N ≤ 1940). Information on phenotype description, sample size and exact heritability and $p$-values is available in Supplementary Data 6. The error bars represent standard errors.

Evidence for GREML $h^2_{SNP}$ estimates was based on likelihood ratio tests. No adjustments for multiple-testing were carried out. Estimates were based on transformed scores: deviance residuals (for categorical phenotypes) or rank-transformed residuals (for continuous phenotypes). DCDQ (Developmental Coordination Disorder Questionnaire), GREML (Genome-based restricted maximum likelihood), $h^2_{SNP}$ (Single nucleotide polymorphism-based heritability), ODD (oppositional defiant disorder), RBSR (Repetitive Behaviour Scale-Revised).

phenotypic homogeneity (as social difficulties are present across the entire ASD spectrum), or little contributions of common genetic variation to phenotypic variation in an ASD case-only sample, or lack of power. These findings corroborate previous analyses in SPARK, where phenotypic factor scores underlying social phenotypes showed little evidence for $h^2_{SNP}$[8]. Next, we identified phenotypes that may share common genetic variation (GREML $r_g$ $p \leq 0.1$, Supplementary Fig. 3) to enable the identification of overarching genetic factors and combined them into genetically related phenotype subsets (Stage II, Supplementary Data 2, Supplementary Note 1, Supplementary Fig. 3). To describe the genomic covariance within each phenotype subset, we developed and implemented a data-driven modelling approach (Stage III, Fig. 1b, Methods). Phenotype subsets with robustly identified genetic structures were eventually combined and a final GRM-SEM model fitted to the data (Supplementary Note 2). Note that in the presence of collinearity problems, genetically highly correlated measures were replaced by a single proxy phenotype (Supplementary Note 1).

In short, our data-driven modelling approach (Fig. 1b, Methods) included a step-wise combination of principal component analysis (PCA), exploratory factor analysis (EFA) and GRM-SEM (Fig. 1b, steps i-vii). Specifically, we estimated the total genomic covariance, as derived

from a saturated GRM-SEM model (Methods, Supplementary Note 2, Supplementary Fig. 4, Supplementary Fig. 5, Supplementary Data 3). From this estimate, we predicted the number of genetic factors (based on PCA eigenvalues) and their underlying genetic structure (based on EFA). Eventually, this information was used to build a hybrid Independent Pathway/Cholesky (IPC) GRM-SEM model, where the structure is only modelled within the genetic part of the data, while the residual part is always fitted to a saturated (Cholesky) model (Methods). IPC models have previously been shown to provide a superior model fit compared to other a priori-defined models[14], which was confirmed as part of sensitivity analysis (shown below, Methods, Table 1, Supplementary Data 3, Supplementary Data 4).

The final and best-fitting GRM-SEM IPC model in SPARK (Table 1), representing all phenotypic subsets, had three independent factors (Fig. 3), corresponding to (1) better language performance ($A_{lang}$); (2) developmental motor delay ($A_{dev}$); and (3) behavioural problems ($A_{beh}$). Jointly these factors covered one core (repetitive behavioural features) and four non-core (language/cognition, general behaviour, developmental, and motor) ASD phenotype domains. The model fit was highly comparable to a saturated model (Table 1, $p_{LRT} > 0.99$) and the model-predicted covariance closely matched the observed phenotypic covariance (SRMR = 0.002). Sensitivity analyses confirmed the

**Table 1 | Model fit comparison**

| Model | Type | Log-likelihood | $N_{par}$ | AIC | BIC | SRMR | LRT$_{Cholesky}$ | | LRT$_{Bi\text{-}factor}$ | |
|---|---|---|---|---|---|---|---|---|---|---|
| | | | | | | | $\Delta\chi^2(\Delta df)$ | $p$ | $\Delta\chi^2(\Delta df)$ | $p$ |
| SPARK, $N_{traits} = 8$, $N_{ind} = 5279$ | | | | | | | | | | |
| Cholesky | saturated | −15248.61 | 72 | 30641.23 | 31114.37 | 0.002 | – | | – | |
| Bi-factor | three-factor | −15249.97 | 62 | 30623.94 | 31031.37 | 0.002 | 2.71(10) | 0.99 | – | |
| IPC best-fit | three-factor | −15250.96 | 53 | **30607.92** | **30956.21** | 0.002 | 4.69(19) | >0.99 | 1.98(9) | >0.99 |
| SSC, $N_{traits} = 8$, $N_{ind} = 1940$ | | | | | | | | | | |
| Cholesky | saturated | −6342.50 | 72 | 12828.99 | 13230.07 | 0.008 | – | | – | |
| Bi-factor | three-factor | −6342.59 | 63 | 12811.18 | 13162.12 | 0.014 | 0.19(9) | >0.99 | – | |
| IPC best-fit | three-factor | −6342.60 | 53 | **12791.19** | **13086.43** | 0.017 | 0.20(19) | >0.99 | 0.01(10) | >0.99 |

The genomic covariance structure across SPARK and SSC phenotypes was modelled using saturated, bi-factor and multi-factor GRM-SEM IPC models (additional comparisons with one-factor IPC models are shown in Supplementary Data 4). The fit across models was compared with likelihood ratio tests (LRT), AIC and BIC. The lowest AIC and BIC values are shown in bold.
*AIC* Akaike information criterion, *BIC* Bayesian information criterion, *IPC* Hybrid Independent Pathway (genetic part)/Cholesky (residual part) model, $N_{par}$ number of parameters, *SRMR* standardised root mean square residual.

independence of factors (bi-factor models, Table 1) and the comparability of estimates with GREML (shown below). To interpret the identified factor structure, we focused on standardised genetic factor loadings accounting for ~10% phenotypic or liability variation ($|\lambda| \geq 0.3$[19]).

The language performance factor ($A_{lang}$) was most strongly related to higher language level ($\lambda_{lang} = 0.46$, SE = 0.08), lower liability to language disorder ($\lambda_{lang} = -0.35$, SE = 0.09) and earlier age of self-feeding ($\lambda_{lang} = -0.38$, SE = 0.14) and accounted for at least half of the trait $h^2_{SNP}$ estimates (50–100%, Supplementary Data 5). Variation in the language performance factor was also positively associated with cognitive functioning as measured by cognitive age level (Supplementary Fig. 5b, e). Notably, the language performance factor also uncovered inverse correlations between children's language ability (e.g. language level) and the age of self-feeding (GRM-SEM $r_g = -0.71$, SE = 0.25, Fig. 3d). The developmental motor delay factor ($A_{dev}$) captured a later age of crawling ($\lambda_{dev} = 0.47$, SE = 0.10), less motor control (DCDQ control during movement, $\lambda_{dev} = -0.33$, SE = 0.13) and more RBSR self-injurious behaviour ($\lambda_{dev} = 0.36$, SE = 0.10), explaining a considerable proportion of genetic variance (44–84% of the $h^2_{SNP}$, Supplementary Data 5). The behavioural problems factor ($A_{beh}$) was linked to RBSR sameness behaviour ($\lambda_{beh} = 0.38$, SE = 0.12) and liability to ODD ($\lambda_{beh} = 0.45$, SE = 0.09) and almost fully explained the $h^2_{SNP}$ of these phenotypes (~100%, Supplementary Data 5).

Specifically, each phenotype had a meaningful genetic factor loading ($|\lambda| > 0.3$) for one factor only. Still, we detected minor genetic heterogeneity for liability to language disorder, with cross-loadings ($p < 0.05$) of all three factors ($\lambda_{lang} = -0.35$, SE = 0.09; $\lambda_{dev} = -0.20$, SE = 0.10; $\lambda_{beh} = -0.20$, SE = 0.10). Given the broad phenotypic definition of developmental language delay and disorder, identified genetic links across independent genetic dimensions may arise due to the broad phenotypic definition capturing multiple underlying aetiologies[20].

Eventually, we compared identified genetic factors in SPARK (Fig. 3) with phenotypic factors identified using an analogous data-driven modelling approach (Methods). In line with Cheverud's conjecture[21], which postulates that phenotypic relationships are likely to be fair estimates of their genetic counterparts, genetic dimensions largely matched corresponding phenotypic dimensions (Supplementary Fig. 6). Nonetheless, several differences between phenotypic and genetic structures became evident, such as for age of self-feeding and RBSR self-injurious behaviour. For example, genetic variation in age of self-feeding was explained by the language performance factor (genetic model: $\lambda_{lang} = -0.38$, SE = 0.14, Fig. 3), while phenotypic variation was accounted for by the developmental motor delay factor (phenotypic model: $\lambda_{dev} = 0.50$, SE = 0.03, Supplementary Fig. 6). Similarly, RBSR self-injurious behaviour was genetically linked to the developmental motor delay factor (genetic model: $\lambda_{dev} = 0.36$, SE =

0.10, Fig. 3), while sharing phenotypic variation with the behavioural problems factor (phenotypic model: $\lambda_{beh} = 0.66$, SE = 0.04, Supplementary Fig. 6). These results leverage the importance of a data-driven genomic covariance modelling approach as genetic relationships may not be fully reflected by phenotypic relationships, given that the latter are also shaped by non-genetic/residual influences.

## Multi-dimensional genetic analyses in simplex ASD

The genetic architecture of ASD is distinctly different in multiplex families with multiple family members with ASD, compared to simplex families with only one child with ASD[22]. ASD liability in simplex families is considerably more often related to de novo mutations[11,23]. Therefore, also common genetic factor structures may differ between exclusively simplex and community ASD samples. To investigate the consistency of latent genetic structures in ASD, we attempted to reproduce the identified best-fitting model from SPARK (Fig. 3) in autistic individuals from SSC simplex families (Fig. 1a, Stage IV). To do so, we selected comparable measures in the SSC (Supplementary Fig. 7, Supplementary Data 6) and applied a data-driven modelling approach (Fig. 1b) to describe the genetic structure.

Matching SSC phenotypes showed little evidence for $h^2_{SNP}$ (Fig. 2b), as expected given the smaller sample size. In particular, both motor (DCDQ scores) and self-injurious behaviour (RBSR) scores had to be excluded from the SSC sample due to convergence problems because of little $h^2_{SNP}$. These two measures were replaced with further language and developmental phenotypes to allow for an empirical identification of three genetic dimensions. As in SPARK, a three-factor model of independent genetic factors fitted the data best (Fig. 4). Sensitivity analyses confirmed the independence of factors (bi-factor models, Table 1, see below). Fit indices indicated a good model fit, comparable to a saturated model (Table 1, $p_{LRT} > 0.99$), and a close match of model-predicted and observed phenotypic covariance (SRMR = 0.017).

The first two genetic factors corresponded to (1) better language performance ($A_{F1}$) and (2) developmental motor delay ($A_{F2}$), matching the SPARK factor structures, $A_{lang}$ and $A_{dev}$, respectively (Fig. 3). In particular, the first genetic factor ($A_{F1}$) accounted for variation in language age level ($\lambda_{F1} = 0.33$, SE = 0.14) and age of self-feeding ($\lambda_{F1} = -0.46$, SE = 0.19), explaining 21–100% of the $h^2_{SNP}$ (Supplementary Data 7). Note, within SPARK, language level (i.e. an individual's everyday language skills) and language age level (i.e. an individual's spoken language for their age level) are strongly correlated (GREML $r_g = 1.00$, SE = 0.24, Supplementary Fig. 3) and showed, when modelled together, similar association patterns. The second genetic factor ($A_{F2}$) described delays in motor development, with the strongest factor loading for age of walking ($\lambda_{F2} = 0.62$, SE = 0.14), capturing up to 93% of $h^2_{SNP}$ (Supplementary Data 7).

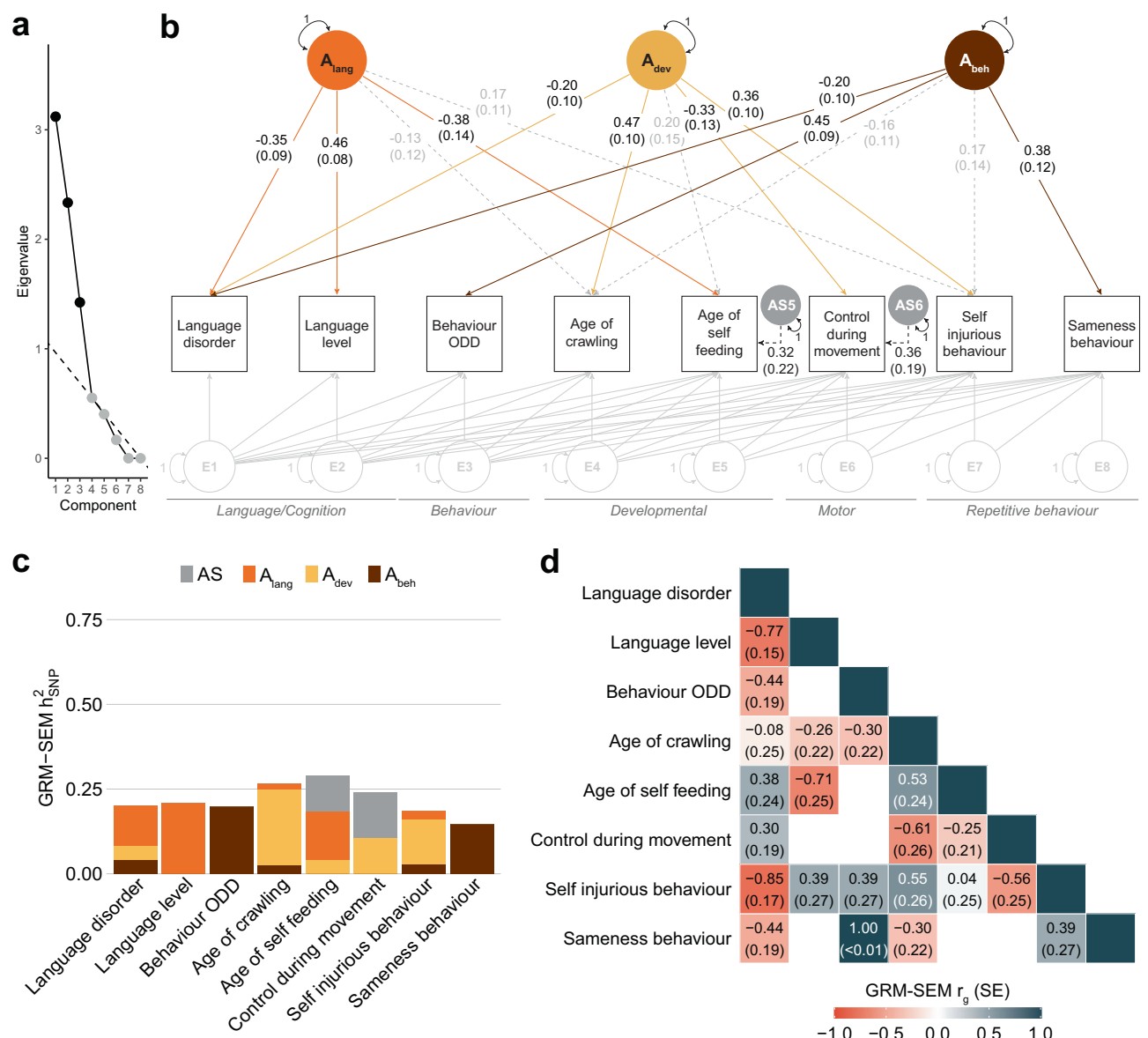

**Fig. 3 | Best-fitting model in SPARK. a** Scree plot based on the eigenvalue decomposition of genetic correlations derived from a GRM-SEM Cholesky model, depicting the number of estimated shared genetic factors (in black) according to an optimal coordinate criterion. The dashed line indicates the "scree" of the plot (grey). **b** Path diagram depicting the best-fitting multi-factor GRM-SEM IPC model. Observed measures are represented by squares and latent variables by circles (A: shared genetic factor, AS: specific genetic factor, E: residual factor). Single-headed arrows define factor loadings (shown with their corresponding SEs). The genetic part of the model has been modelled using an Independent Pathway model. Grey dotted and coloured solid arrows define shared genetic factor loadings with $p > 0.05$ and $p \leq 0.05$, respectively. Black dotted lines define specific genetic factor loadings with $p > 0.05$. The residual part has been modelled using a Cholesky model and all residual factor loadings are shown in grey. The full parameter table is shown

in Supplementary Data 5. Evidence for GRM-SEM factor loadings was assessed with Wald tests (two-sided). Given the multivariate design, no adjustment for multiple comparisons was carried out. **c** Corresponding standardised genetic variance (GRM-SEM $h^2_{SNP}$) plot. SEs for GRM-SEM $h^2_{SNP}$ contributions have been omitted for clarity. **d** Corresponding correlogram of genetic correlations ($r_g$). Numeric values for genetic correlations that are not predicted by the genetic model structure were omitted. $A_{lang}$ (genetic language performance factor), $A_{dev}$ (genetic developmental motor delay factor), $A_{beh}$ (genetic behavioural-problems factor), DCDQ (Developmental Coordination Disorder Questionnaire), $h^2_{SNP}$ (Single nucleotide polymorphism-based heritability), IPC (Independent Pathway-Cholesky GRM-SEM model), ODD (Oppositional Defiant Disorder), RBSR (Repetitive Behaviours Scale-Revised).

The third genetic factor in SSC ($A_{F3}$) showed a different structure compared to SPARK (Fig. 4b). This factor ($A_{F3}$) explained shared genetic variation across language performance and repetitive (RBSR sameness) behaviour, capturing the majority of their $h^2_{SNP}$ (75–100%, Supplementary Data 7). The strongest factor loadings were observed for language age level ($\lambda_{F3} = 0.61$, SE = 0.10), language disorder ($\lambda_{F3} = -0.51$, SE = 0.11), language level ($\lambda_{F3} = 0.37$, SE = 0.07), but also RBSR sameness behaviour ($\lambda_{F3} = 0.51$, SE = 0.12). This cross-trait genetic dimension in the SSC accounted for strong positive genetic

correlations between language and repetitive behaviour (e.g. language level, RBSR sameness behaviour: GRM-SEM $r_g = 0.97$, SE = 0.07, Fig. 4d) which were absent in SPARK (language level, RBSR sameness behaviour: GRM-SEM $r_g = 0$, Fig. 3d). A comparison of genetic and phenotypic factor structures was not possible in the SSC, as the phenotypic model was empirically unidentified using a split-half data-driven modelling approach (Methods) due to convergence problems.

Overall, the consistency of findings in SPARK and the SSC implied a high level of reproducibility of genetic factor structures across

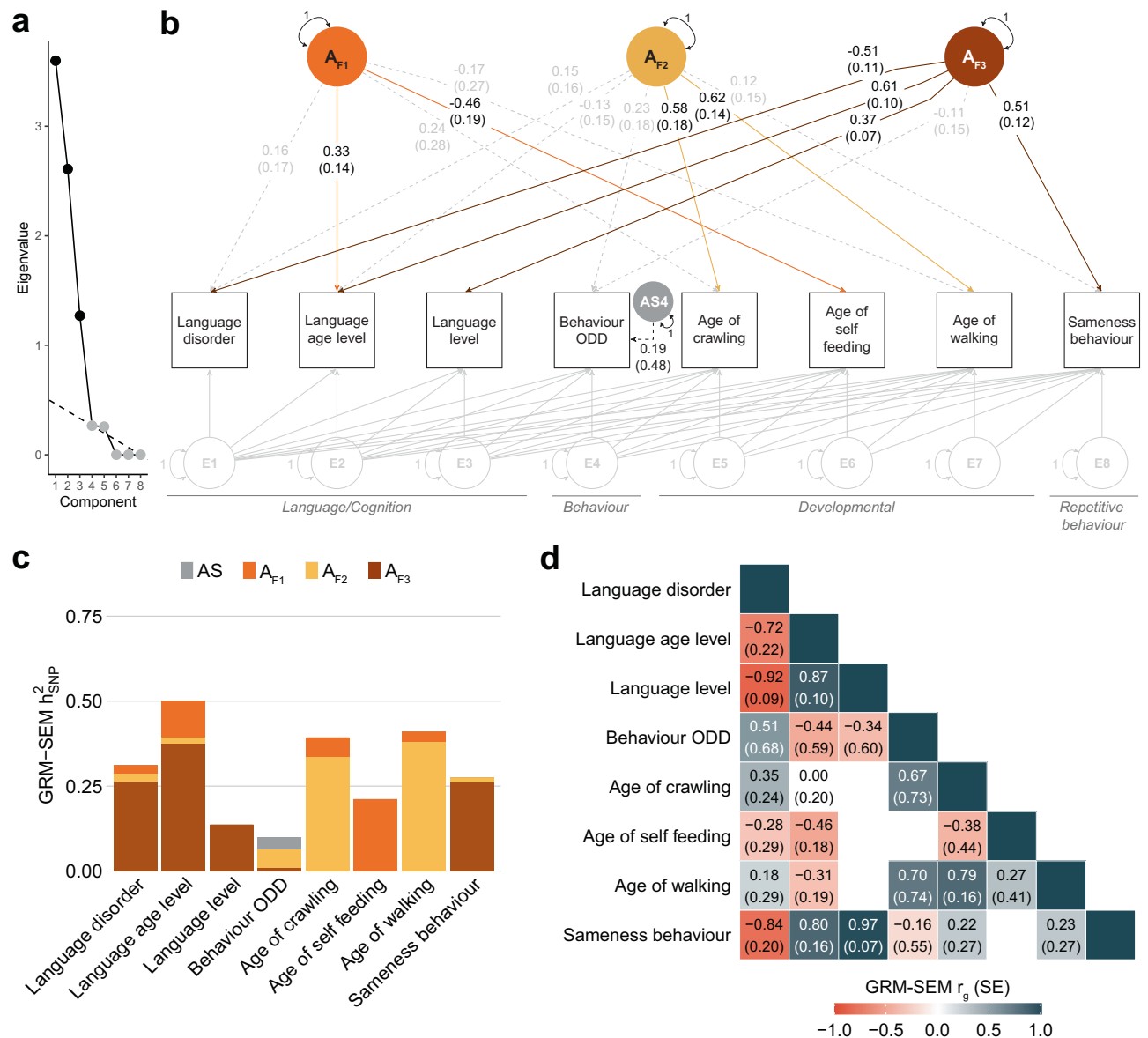

**Fig. 4 | Follow-up multi-factor GRM-SEM model in the SSC. a** Scree plot based on the eigenvalue decomposition of genetic correlations derived from a GRM-SEM Cholesky model, depicting the number of estimated shared genetic factors (in black) according to an optimal coordinate criterion. The dashed line indicates the "scree" of the plot (grey). **b** Path diagram depicting the best-fitting multi-factor GRM-SEM IPC model based on largely comparable phenotypes as studied in SPARK. Observed measures are represented by squares and latent variables by circles (AF: shared genetic factor, AS: specific genetic factor, E: residual factor). Single-headed arrows define factor loadings (shown with their corresponding SEs). The genetic part of the model has been modelled using an Independent Pathway model. Grey dotted and coloured solid arrows define shared genetic factor loadings with $p > 0.05$ and $p \leq 0.05$, respectively. Black dotted lines define specific genetic factor loadings with $p > 0.05$. The residual part has been modelled using a Cholesky model and all residual factor loadings are shown in grey. The full parameter table is shown in Supplementary Data 7. Evidence for GRM-SEM factor loadings was assessed with Wald tests (two-sided). Given the multivariate design, no adjustment for multiple comparisons was carried out. **c** Corresponding standardised genetic variance (GRM-SEM $h^2_{SNP}$) plot. SEs for GRM-SEM $h^2_{SNP}$ contributions have been omitted for clarity. **d** Corresponding correlogram of genetic correlations ($r_g$). Numeric values for genetic correlations that are not predicted by the genetic model structure were omitted. $A_{F1,2,3}$ (Genetic factor 1,2,3), $h^2_{SNP}$ (Single nucleotide polymorphism-based heritability), IPC (Independent Pathway-Cholesky GRM-SEM model), ODD (Oppositional Defiant Disorder).

distinct genetic architectures in ASD, especially for the language performance and developmental motor domains.

## Characterisation of identified genetic factor structures

To enhance the interpretability of identified genetic structures in SPARK, we mapped further variables onto the genetic model structure (Methods). Specifically, we investigated the association between the identified factors and (i) liability to Asperger (compared to other ASD subcategories) (Fig. 5a–c) and (ii) PGS for educational attainment (PGS$_{EA}$) (Fig. 5d–f). ASD subcategory information (DSM-IV-based) can

provide a clinical reference to account for different phenotypic presentations in ASD. Here, it can guide the interpretation of identified genetic dimensions, as genetic liability to Asperger presents a form of autism without significant impairments in language and cognitive development[24]. PGS$_{EA}$ presents a genetic correlate of cognitive functioning[25], but also socio-economic status, including non-cognitive factors such as health and longevity[26]. Given low $h^2_{SNP}$ in the SSC (Fig. 2b), analyses were restricted to SPARK only.

For the best-fitting model in SPARK, liability to Asperger was genetically associated with the language performance factor (Fig. 5a,

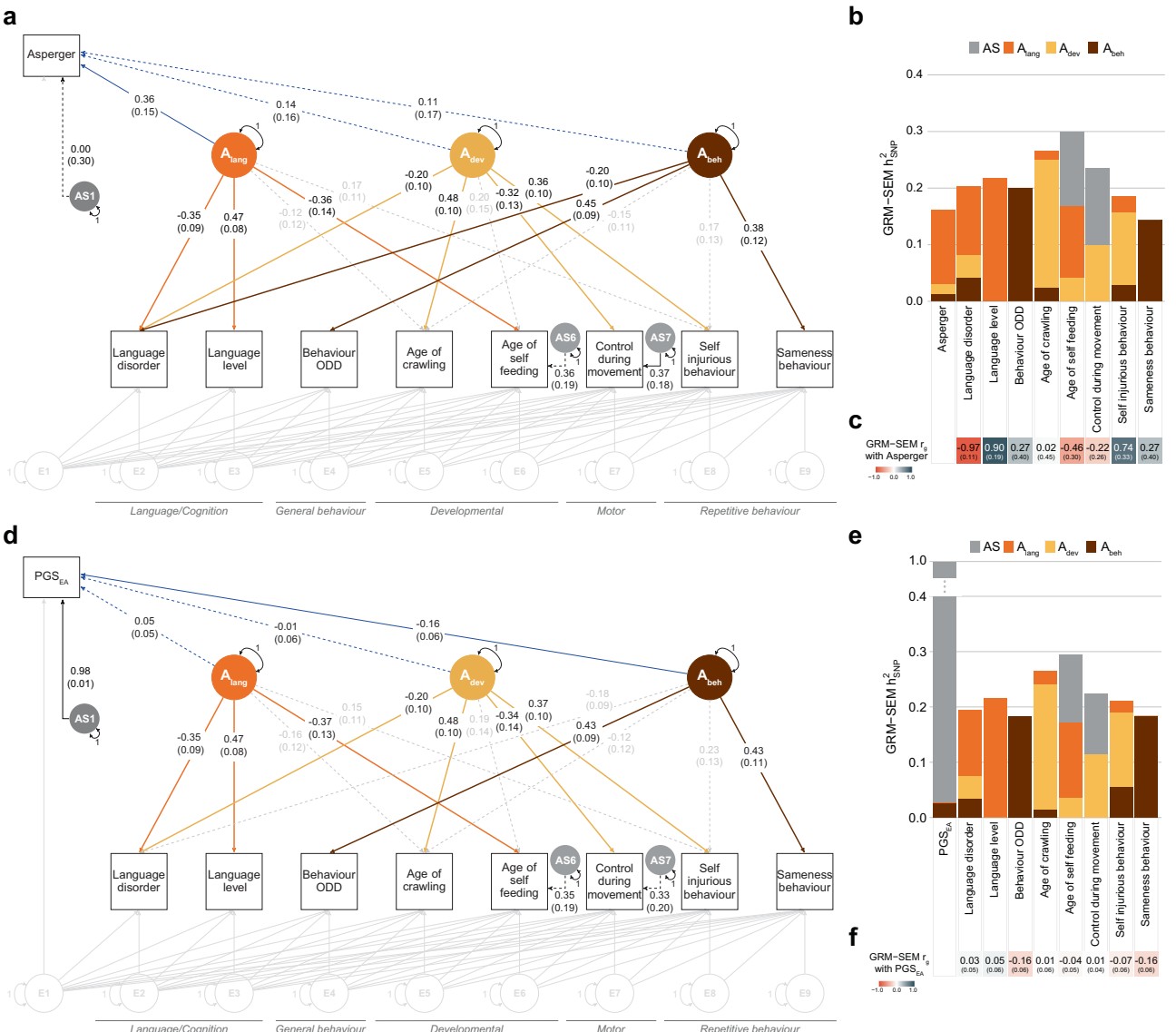

**Fig. 5 | Characterisation of identified genetic factor structures in SPARK. a** Path diagram of an extended GRM-SEM IPC model mapping liability to Asperger (reference: Asperger against other ASD subcategories) onto the model structure of the best-fitting model in SPARK. **b** Corresponding standardised genetic variance (GRM-SEM $h^2_{SNP}$) plot. SEs for GRM-SEM $h^2_{SNP}$ contributions have been omitted for clarity. **c** Genetic correlations with liability to Asperger. **d** Path diagram of an extended GRM-SEM IPC model mapping polygenic scores for educational attainment (PGS$_{EA}$) onto the model structure of the best-fitting model in SPARK. **e** Corresponding standardised genetic variance (GRM-SEM $h^2_{SNP}$) plot. SEs for GRM-SEM $h^2_{SNP}$ contributions have been omitted for clarity. **f** Genetic correlations with PGS$_{EA}$. **a**, **d** Observed measures are represented by squares and latent variables by circles ($A_{lang}$/$A_{dev}$/$A_{beh}$: shared genetic factor, AS: specific genetic factor, E: residual factor). Single-headed arrows define factor loadings (shown with their

corresponding SEs). The genetic part of the model has been modelled using an Independent Pathway model. Grey dotted and coloured solid arrows define shared genetic factor loadings with $p > 0.05$ and $p \leq 0.05$, respectively. Black dotted lines define specific genetic factor loadings with $p > 0.05$. Factor loadings for the mapping variable are shown in blue (dotted: $p > 0.05$; solid $p \leq 0.05$). The residual part has been modelled using a Cholesky model (grey). Evidence for GRM-SEM factor loadings was assessed with Wald tests (two-sided). Given the multivariate design, no adjustment for multiple comparisons was carried out. $A_{lang}$ (genetic language performance factor), $A_{dev}$ (genetic developmental motor delay factor), $A_{beh}$ (genetic behavioural-problems factor), DCDQ (Developmental Coordination Disorder Questionnaire), $h^2_{SNP}$ (single nucleotide polymorphism-based heritability), IPC (Independent Pathway-Cholesky GRM-SEM model), ODD (Oppositional Defiant Disorder), RBSR (Repetitive Behaviours Scale-Revised), $r_g$ (genetic correlation).

$\lambda_{lang} = 0.36$, SE = 0.15). Genetic correlations between liability to Asperger and language level (Fig. 5c, GRM-SEM $r_g = 0.90$, SE = 0.19) were positive, consistent with the absence of language problems in this ASD subcategory[3]. In contrast, PGS$_{EA}$ were associated with reduced behavioural problems (Fig. 5d, $\lambda_{beh} = -0.16$, SE = 0.06), conditional on the language performance and developmental motor delay dimensions. Consistent with previous research[8,9], genetic correlations of PGS$_{EA}$ with behavioural measures such as sameness behaviour were inverse (Fig. 5f, GRM-SEM $r_g = -0.16$, SE = 0.06), strengthening support for links with repetitive behaviour.

## Sensitivity analyses

We carried out a series of sensitivity analyses. Across both cohorts, we, first, confirmed the independence of identified genetic factors by comparing the best-fitting GRM-SEM IPC models with GRM-SEM bi-factor models (Methods), which showed a similar fit and model structure ($p_{LRT} \geq 0.94$, Table 1, Supplementary Fig. 8, Supplementary Fig. 9). Second, we corroborated GRM-SEM predicted $h^2_{SNP}$ (Supplementary Fig. 10) and $r_g$ (Supplementary Fig. 11) estimates for the best-fitting GRM-SEM IPC models through comparisons with corresponding GREML estimates that showed consistent 95%-confidence intervals.

Note that GREML $r_g$ estimates could not be fitted in the SSC given low $h^2_{SNP}$. Third, we illustrated the superiority in model fit for the best-fitting GRM-SEM IPC models by comparing their fit with exploratory GRM-SEM models (Supplementary Data 4), such as one-factor independent pathway and one-factor IPC models (Methods, Supplementary Fig. 12). Fourth, we validated the predictive value of the implemented data-driven genomic covariance modelling approach. Specifically, we demonstrated the interchangeability of different EFA algorithms (Supplementary Data 8) and the strong correlation between EFA-predicted and GRM-SEM-predicted (best-fitting model) factor loadings (SPARK: Pearson $r > 0.986$, SSC: Pearson $r > 0.992$, Supplementary Data 9, Supplementary Fig. 13). Lastly, to exemplify the robustness of our modelling approach, we performed proof-of-principle simulations (Supplementary Note 3) that exhibited sufficient 95%-confidence interval estimate coverage and provided little evidence for bias (Supplementary Data 10–13, Supplementary Figs. 14–15). Together, these analyses demonstrate that multivariate genetic methods are required to accurately depict the common genetic architecture in ASD and, potentially, other complex traits. Our findings emphasise that simplistic a priori-defined models are insufficient to capture the complexity of genetic effects.

## Discussion

Investigating genomic covariance across a broad spectrum of phenotypes in ASD using SEM-based techniques, this case-only study of two large autism cohorts demonstrates that the common genetic architecture of ASD is multi-dimensional. Here, we identified evidence for at least three independent common genetic dimensions associated with phenotypic heterogeneity in ASD.

For SPARK, a community ASD sample, we identified three common genetic factors explaining predominantly variation in language performance, developmental motor delay and behavioural problems, respectively. Within the SSC, a sample of simplex-only families, we uncovered structural similarities, indicating conceptual replication. Hence, our findings not only strengthen the evidence for common genetic contributions to phenotypic variation in ASD[8,9,12], but also offer insight into the multi-dimensional genetic architecture. Specifically, we show that the majority of $h^2_{SNP}$ in ASD-only samples can be explained by shared genetic factors, for most phenotypes investigated. The major difference across cohorts concerned the genetic relationship between language and behavioural phenotypes. While genetic factors of language performance and behaviour were unrelated in a community ASD sample (SPARK), the underlying phenotypes were strongly genetically linked in simplex ASD (SSC) and captured by a single dimension, suggesting ascertainment-specific association patterns.

Across both cohorts, we found evidence for an independent language performance factor, as validated through association with higher liability to Asperger in SPARK. Although an Asperger ASD subcategory diagnosis is not included in the DSM-5 anymore, our findings confirm that autistic individuals differ considerably in their language presentation[27]. While some children with ASD reach intact structural language skills, others are delayed or never master functional spoken language[27]. Here, our analyses uncovered genomic covariance between (higher) language level and (earlier) age of self-feeding with a spoon, an important personal-social developmental milestone which typically developing children master at about 15–18 months[28,29]. Notably, the genetic influences contributing to the age by which children self-feed with a spoon were distinct from genetic factors underlying other motor developmental achievements, such as crawling, sitting or walking, when studied in SPARK. Infant autonomy in feeding, especially eating with the family, has been related to more advanced child language production and comprehension[30]. Especially within SPARK, age of self-feeding with a spoon showed moderate to strong relationships with multiple language-related phenotypes and may

present an early marker of cognitive and language development in ASD. As the language performance factor captured, by proxy, also variation in cognitive age level, as shown for a smaller phenotypic subset, it is likely that this genetic factor is representative of a joint language/cognitive domain.

We also found robust evidence for a genetic factor that is related to developmental motor delay within SPARK and the SSC, explaining genetic variation underlying growth, such as the age of crawling, a developmental milestone children typically master between 9 and 18 months of age[31]. Within SPARK, genetic variation contributing to the age of crawling (a proxy of the age of walking and sitting) was also shared with DCDQ motor control during movement (a proxy of DCDQ total score and fine motor handwriting), language disorder and RBSR self-injurious behaviour. These findings support the contribution of common genetic influences to variation in motor abilities, beyond association with de novo mutations[9]. The spectrum of genetically linked developmental phenotypes, furthermore, extends reports of genetic associations between ASD polygenic liability and later age of walking in population-based samples[32].

Genetic relationships between language/cognition phenotypes and behaviour across cohorts were heterogeneous, highlighting ascertainment-specific patterns. Within SPARK, the behavioural genetic dimension was independent of the language performance dimension. The behavioural-problems factor explained liability to ODD and variation in repetitive behaviour, especially RBSR sameness behaviour, which is a proxy of RBSR total scores and ritualistic behaviour, but not self-injurious behaviour. This finding matches previously reported distinct phenotypic factor structures between self-injurious and other types of repetitive behaviours[8]. Consequently, self-injurious actions may, at least partially, be aetiologically distinct from other forms of repetitive behaviour. As in previous research adopting a case-only design[8,9], the behavioural problems factor was inversely associated with PGS$_{EA}$ in SPARK. Our analyses demonstrated that this association can be observed conditional on genetic links with the language performance or the developmental motor delay factor, neither of which were related to PGS$_{EA}$. Thus, our findings suggest that behavioural problems within a community ASD sample vary primarily with non-cognitive correlates of socio-economic status. However, it is important to highlight that a large proportion of the PGS$_{EA}$ genetic effects are not due to direct effects, but indirect effects (e.g. non-transmitted parental genetic influences), other forms of gene-environment correlation or assortative mating[33]. Therefore, the nature and causality of PGS$_{EA}$ associations cannot be determined from our analysis.

In contrast, within the SSC, we observed substantial genetic overlap between most language-related phenotypes and RBSR sameness behaviour. Simplex ASD, compared to multiplex ASD, is more often related to de novo mutations[11,23]. Our findings may, therefore, present aetiological differences unique to simplex ASD, consistent with qualitative differences in the common genetic architecture of ASD individuals carrying de novo variants[5,6]. Alternatively, genetic links between behaviour and language phenotypes in the SSC might, to some degree, be a consequence of collider bias[34]. Simplex families are recruited following strict ascertainment schemes[16]. Collider bias can arise when two measures, such as behaviour and language/cognition, are independently related to a third variable, such as common genetic variation, and that third variable is conditioned upon[34]. Here, the preferential ascertainment of simplex families depleted for inherited genetic variation[35], including common variation, may introduce artificial genetic relationships between behaviour and language/cognition.

Our study has multiple strengths, but also limitations. First, we developed a data-driven modelling approach that utilises directly genotyped genome-wide information and facilitates building accurate multi-dimensional models of genomic covariance without the need for summary statistics or the prediction of structure through phenotypic

SEM. Although phenotypic structures were, overall, fair estimates of genetic structures[21], they could not fully capture underlying genetic relationships. Second, our multivariate analysis approach allowed for the detection of multiple genetic dimensions in ASD, providing sufficient statistical identifiability (i.e. degrees of freedom). Third, we mapped external references, such as liability to Asperger and $PGS_{EA}$, onto genetic relationships observed within individuals of ASD to aid the interpretation of genetic structures across different research designs. However, GRM-SEM models rely, fourth, on population-based assumptions (e.g. Hardy-Weinberg equilibrium of genotype distributions) and we may have excluded individuals or genetic variation that do not meet these expectations. Fifth, we studied transformed scores to aid model simplicity and model convergence and, therefore, cannot exclude bias. However, it is unlikely that data transformations have profoundly changed underlying genetic relationships, given the robustness of sensitivity analyses and the consistency of results with previous findings. Sixth, our study cannot yet address sex-specific differences in common genetic architectures, as previously reported[9], especially across non-European ancestry backgrounds. Because the prevalence of ASD is higher in males, the sex distribution in both SPARK and SSC is skewed and our results may, therefore, be less generalisable for females. Similarly, studying subgroups of individuals with and without ID was not yet feasible, given limited power (e.g. based on simple GREML models, the power for subgroup analyses with $N \sim 1200$ and $h^2_{SNP}$ of 0.2, as observed in this study, is 0.11). Stratifying GRM-SEM models across common, rare and de novo carriers in sufficiently powered samples, accounting for differences between males and females, will shed further light onto the complex links between genetic and phenotypic heterogeneity in ASD, as part of future studies.

Together our results describe phenotypic variation in ASD as complex traits that are, at least partially, genetically linked due to common genetic factors that are augmented by ascertainment-specific patterns. More generally, the implementation of a data-driven genomic covariance modelling approach demonstrates that multidimensional common genetic architectures can be accurately identified using direct genome-wide genotyping data.

## Language choice
We are aware that the choice of language plays an important role in the autism community[36,37]. While some individuals prefer person-first language (i.e. individuals with autism), others prefer identity-first language (i.e. autistic individuals). Based on the preferences of the SPARK community[38], we have used person-first ("individuals with autism" or "individuals with ASD") and identity-first ("autistic individuals" or "ASD individuals") terms interchangeably. We acknowledge and respect each individual's preference to identify themselves.

## Methods
### Samples
The SPARK cohort (https://sparkforautism.org/)[15] is a nationwide autism study across the United States including simplex and multiplex families. We studied SPARK phenotype (SPARK Collection Version 3) and genome-wide (SPARK 30 K release) data. This data freeze includes 59,218 individuals between ages 1 and 85, who received a professional diagnosis of ASD/autism (85% < 18 years; 79% male), their biological parents, and, if available, one unaffected control sibling as well as all autistic siblings for multiplex families (21,689 trios (including simplex families); 6552 multiplex families). Written informed consent was completed by the parent or legal guardian of the children participating in the study.

The SSC cohort (https://www.sfari.org/resource/simons-simplex-collection/)[16] is a collection of simplex families from the United States. We investigated phenotype (version 15.3) and genome-wide (whole-genome 2 data release) data. This data freeze represents 2591 autistic children aged 4 to 17 years 11 months, including 2643 simplex families

with one (and only one) child with ASD and their unaffected biological parents and unaffected siblings. Informed consent and assent were provided for all participants.

We received ethical approval to access and analyse pre-collected de-identified genotype and phenotype data from these cohorts from the Radboud University Ethics Committee Social Science. All analyses were restricted to individuals with ASD with phenotypic and genetic information.

### Genotype information
**SPARK.** Genotyping was performed using the Infinium Global Screening Array-24 v.1.0. After individual and variant quality control (QC), we included in this study 5331 unrelated individuals (79.85% males, median age: 9 years) of European ancestry, with an ASD diagnosis, and with genetic and phenotype information available (Supplementary Fig. 1, Supplementary Methods 1). Individuals were excluded due to confirmed genetic syndromes/conditions, birth complications (i.e. birth defects, foetal alcohol syndrome, bleeding into the brain, insufficient oxygen at birth), other cognitive impairments or a brain injury (i.e. brain infection, lead poisoning, traumatic brain injury). A GRM[17] based on directly genotyped markers ($N_{SNPs}$ = 450,491) was created in PLINK (v1.9)[39], applying a relationship cut-off of 0.05.

**SSC.** Genotyping was performed using three arrays: Illumina Human1M v1.0, Illumina Human1M-Duov3 and Illumina HumanOmni2.5. For each array, individual and variant QC was performed separately (Supplementary Methods 2). Subsequently, genotype data were merged across the three arrays and again subjected to individual and variant-based QC. After QC, 1946 unrelated individuals (86.33% males, median age: 9 years) of European ancestry, diagnosed with ASD, and with genetic and phenotype information were included in the study (Supplementary Fig. 7). Individuals were excluded because of premature birth, brain injury/damage/abnormality, prenatal/birth complications, confirmed genetic syndromes/conditions, severe sensory/motor difficulties or nutritional/psychological deprivation. A GRM[17] based on directly genotyped markers ($N_{SNPs}$ = 457,961) was created in PLINK (v1.9)[39], applying a relationship cut-off of 0.05.

### Phenotypes
**SPARK.** We studied 47 parent-reported measures of ASD phenotypes and co-morbid disorders/disabilities spanning the domains of language and cognition (nine measures), general behaviour (nine measures), repetitive behaviour (seven measures), social (two measures) and motor abilities (six measures), as well as affective disorders (three measures) and developmental milestones (11 measures). Phenotypes were extracted from the Basic Medical Screening Questionnaire (BMS), the Social Communication Questionnaire-Lifetime (SCQ)[40], the SPARK Background History Questionnaire (BGHX), the Repetitive Behaviours Scale-Revised (RBSR)[41], and the Developmental Coordination Disorder Questionnaire (DCDQ)[42] (Supplementary Fig. 1, Supplementary Methods 1). The selected phenotypes included 21 categorical (within-sample prevalence of at least 5%) and 26 continuous phenotypes. At least 2910 autistic individuals had phenotype and genotype data per trait (Supplementary Data 1). Among all the studied individuals in the SPARK sample, information on ASD subcategories was available for only 1754 individuals: Asperger ($N_{ind}$ = 716, 79.05% males, age range: 2–60 years), childhood autism ($N_{ind}$ = 624, 81.57% males, age range: 1–55 years) and Pervasive Developmental Disorder Not Otherwise Specified (PDD-NOS, $N_{ind}$ = 414, males=80.67% males, age range: 2–45 years).

**SSC.** For follow-up analyses, we studied 17 parent-reported measures of language and cognition (five measures), general behaviour (one measure), repetitive behaviour (four measures), and motor abilities (three measures), as well as developmental milestones (four measures) that were comparable to SPARK measures. Phenotypes were selected

from the SSC BGHX, the SSC Diagnosis Summary Form, the SSC Medical History Interview, RBSR[41], DCDQ[42], the Child Behavior Checklist (CBCL 6–18)[43], and the Autism Diagnostic Observation Scale (ADOS)[44] (Supplementary Fig. 7, Supplementary Methods 2). The selected phenotypes included three categorical (within-sample prevalence of at least 5%) and 14 continuous measures, and at least 1449 autistic individuals had phenotype and genotype data (Supplementary Data 6).

## Phenotype transformations

To adjust for covariates, all phenotypes were regressed against sex, age, age squared, and ten ancestry-informative principal components[45], where the latter corrected for subtle ancestry differences among individuals of Caucasian ancestry. This was performed using ordinary least square regression for continuous variables and binary logistic regression for categorical variables.

**Categorical variables.** After adjusting for covariates, deviance residuals were constructed by extracting the logistic model residuals using the *resid* function (*R:stats* v4.0.2). Deviance residuals are computed as the difference between the logistic model fit to the data against the fit of a saturated model.

**Continuous variables.** After adjusting for covariates, model residuals were rank-transformed and regressed again on covariates to achieve normality of transformed scores without a re-introduction of covariate effects (fully-adjusted two-stage rank normalisation)[46].

To ensure the validity of the transformed scores, we carried out extensive sensitivity analyses. For this purpose, we compared GREML $h^2_{SNP}$ estimations (Supplementary Fig. 2, Supplementary Fig. 16) and phenotypic correlations for untransformed and transformed scores (Supplementary Fig. 17, Supplementary Fig. 18). Pertinent to this work, analyses were conducted with transformed scores to ease the modelling process, i.e. deviance residuals for categorical phenotypes and rank-transformed scores for continuous phenotypes.

## Study design

We developed a multi-stage SEM-based modelling design to identify and characterise shared genetic factor structures (Fig. 1a).

**Stage I. Univariate and bivariate genetic variance analyses in SPARK.** Within stage I, we screened for, by trend, heritable and genetically interrelated clusters of phenotypes across the heterogenous genetic spectrum in SPARK, facilitating model-building convergence. Univariate ($h^2_{SNP}$) and bivariate ($r_g$) genetic variance analyses were carried out with GREML using Genome-wide Complex Trait Analysis (GCTA)[17] (see below).

**Stage II. Phenotype selection in SPARK.** Phenotype subsets were identified based on genetic correlations (GREML $r_g$, $p \leq 0.1$). Phenotype subsets were selected to successively construct a comprehensive GRM-SEM model (Supplementary Note 1, Supplementary Data 2). We adopted this strategy as GRM-SEM models are computationally expensive[13]. For example, a Cholesky decomposition model for 8 traits, as fitted within this study, can require up to 6 weeks of computing time even on a system incorporating at least four parallel cores of 3 GHz, and requiring up to 40 Gb (max vmem) memory. Therefore, building a model from smaller phenotype subsets ensures the robustness of identified structures and reduces the computational burden. Where item scales of the same instrument were genetically redundant (GRM-SEM $r_g = 1$), we retained a single representative measure (or proxy) only (Supplementary Note 1, Supplementary Fig. 4) to avoid collinearity that can affect model convergence. The selection of proxies for measures of the same questionnaire (i.e. BGHX, DCDQ and RBSR) was guided by uni-dimensional GRM-SEM models (Supplementary Note 1, Supplementary Fig. 4).

**Stage III. Multivariate modelling of genomic covariance in SPARK.** As part of stage III, we aimed to identify the best-fitting multi-dimensional GRM-SEM models for the selected phenotype subsets and, eventually, a combined set of measures (Supplementary Note 2). We fitted a series of GRM-SEM saturated (Cholesky) models, PCA models, EFA models, and, finally, GRM-SEM multi-factor models as well as a priori-defined GRM-SEM models. Our step-wise data-driven genomic covariance modelling approach (Fig. 1b) is described in detail below.

**Stage IV. Multivariate modelling of genomic covariance in SSC.** During Stage IV, the best-fitting SPARK model was followed-up in autistic individuals from simplex families (SSC).

## Univariate and bivariate genetic variance analyses

Univariate ($h^2_{SNP}$) and bivariate ($r_g$) analyses were carried out with GREML[18], as implemented in GCTA (v1.93) software[17]. Note that $h^2_{SNP}$ reflects the proportion of phenotypic variance among autistic individuals as explained by genotyped variants (SNPs) and $r_g$ reflects the extent to which two phenotypes are influenced by the same genetic factors. GRMs were constructed from genome-wide genotyping information (Supplementary Methods 1, Supplementary Methods 2).

## Genetic relationship matrix structural equation modelling

We modelled the multivariate genetic variance structure of ASD phenotypes using GRM-SEM as implemented in *grmsem* (*R:grmsem*, v1.1.2, https://gitlab.gwdg.de/beate.stpourcain/grmsem) previously known as *gsem*[13,14].

GRM-SEM applies structural equation modelling techniques to analyse genomic covariance in samples of unrelated individuals using a maximum likelihood approach[13]. We define a multivariate normal phenotype Y (for 1…k traits), where each individual $i$ follows $Y_i \sim N_k$ (μ, $\Sigma_V$). We define $\Sigma_V$, the expected phenotypic variance of Y, as the sum of the expected genetic and residual variance components, $\Sigma_A$ and $\Sigma_E$:

$$\Sigma_V = \Sigma_A + \Sigma_E \tag{1}$$

where $\Sigma_V$, $\Sigma_A$ and $\Sigma_E$ are symmetric $k \times k$ matrices. The residual variance component, potentially, includes environmental factors, random error, non-additive genetic variance, rare variance or any other genetic influence not captured by the GRM[13,14,17]. Within GRM-SEM, genetic and environmental influences are modelled as latent variables. The phenotypic variance for each measure Y can be dissected into genetic and residual influences (AE model), analogous to twin research[47]:

$$\Sigma_V = \Lambda_A \Psi_A \Lambda_A^T \otimes G + \Lambda_E \Psi_E \Lambda_E^T \otimes I \tag{2}$$

where $\Lambda_A$ and $\Lambda_E$ are matrices of genetic and residual factor loadings with dimensions $k \times p$, where $p$ is the number of factor loadings. $\Psi_A$ and $\Psi_E$ are $p \times p$ matrices of genetic and residual factor variances, respectively. G is a $n \times n$ GRM matrix for all pairs of $n$ independent individuals[17] constructed from the variants presented on a genome-wide genotyping chip, and I is a $n \times n$ identity matrix. The symbol $\otimes$ denotes the Kronecker product. Thus, we assume besides structured genetic covariance also structured residual covariance that can contribute to phenotypic covariance patterns[14]. In this work, $\Psi_A$ and $\Psi_E$ have been restricted to an identity matrix, given modest EFA-predicted genetic correlations between latent variables (Supplementary Data 8, Supplementary Note 2). Bi-factor models were fitted to confirm the independence of genetic factor structures (see below). We, furthermore, assume that common genetic variance in ASD individuals can be modelled according to population-based principles and that by expressing the phenotype of each individual $i$ as a deviation from the mean (Z scores), the estimation of means can be omitted.

We fitted the following multivariate models[13,14] as implemented into our multi-stage modelling design (Fig. 1, see below):

i. *Cholesky model*: The Cholesky decomposition model (Supplementary Fig. 12a) is a saturated i.e. fully parametrised descriptive model without any restrictions on the structure of latent genetic and residual influences. This model is fitted to the data through the decomposition of both the genetic variance and residual variance into as many latent variables (factors) as there are observed variables. Here, $\Lambda_A$ and $\Lambda_E$ are $k \times k$ lower diagonal matrices. Note that other saturated models, such as direct symmetric models[48], were not fitted due to convergence problems with multicollinear data (not shown).

ii. *Independent pathway model*: The independent pathway model (Supplementary Fig. 12b) specifies one or more shared genetic and one or more shared residual factors, where $n_{AC}$ is the number of shared genetic factors and $n_{EC}$ is the number of residual factors, in addition to trait-specific genetic and residual influences, one for each trait. $\Lambda_A$ and $\Lambda_E$ have the dimensions $k \times p_a$ and $k \times p_e$, respectively, where $p_a$ is the sum of $n_{AC} + k$, and $p_e$ is the sum of $n_{EC} + k$. Pertinent to this study, we fitted one-factor models only ($n_{AC} = n_{EC} = 1$).

iii. *Hybrid Independent Pathway/Cholesky model (IPC)*. The IPC model (Supplementary Fig. 12c) structures the genetic variance as an independent pathway model (consisting of shared and measurement-specific influences where $\Lambda_A$ has a dimension of $k \times (n_{AC} + k)$) and the residual variance as a Cholesky model (where $\Lambda_E$ is a lower diagonal $k \times k$ matrix). In this study, we fitted one-factor ($n_{AC} = 1$; $k_{traits} \geq 3$) and multi-factor ($n_{AC} = 2$, $k \geq 6$; $n_{AC} = 3$; $k \geq 8$) IPC models.

iv. *Bi-factor IPC model*. The bi-factor model[49] consists of a general factor and one or more grouping factors, where each trait loads on the general factor, assuming statistical independence between these latent genetic dimensions. Given the bi-factor parametrisation, the model benefits from rotational invariance and unlimited dimensionality[50].

The relative goodness-of-fit for each model was evaluated with likelihood ratio tests (LRTs), the Akaike information criterion (AIC) and the Bayesian information criterion (BIC)[51]. The absolute goodness-of-fit was assessed with the standardised root mean square residual (SRMR)[52], as the standardised difference between the observed and predicted correlation, accounting for the degrees of freedom in GRM-SEM. SRMR values below a cut-off value of 0.08 indicate a good model fit[52].

Evidence for GRM-SEM factor loadings was assessed using Wald tests, based on unstandardised scores, while reported coefficients $\lambda$ represent standardised factor loadings (setting the phenotypic variance to unit variance).

For the best-fitting GRM-SEM models, we estimated heritability ($h^2_{SNP}$), genetic correlations ($r_g$), and factorial co-heritabilities ($f^2_g$, i.e. the proportion of total trait genetic variance explained by a specific genetic factor). We defined bivariate genetic correlation between traits, measuring the extent to which two traits share genetic factors (ranging from −1 to 1)[53] according to

$$r_g = \frac{\sigma_{g12}}{\sqrt{\sigma^2_{g1} \sigma^2_{g2}}} \tag{3}$$

where $\sigma_{g12}$ is the genetic covariance between two traits 1 and 2, and $\sigma^2_{g1}$ and $\sigma^2_{g2}$ are their respective genetic variances. In addition, we estimate the factorial co-heritability $f^2_g$ as the relative contribution of a genetic factor to the genetic variance of a trait, defined as:

$$f^2_g = \frac{\sigma^2_{g_{jt}}}{\sum \sigma^2_{g_{jt}}} = \frac{\sigma^2_{g_{jt}}}{\sigma^2_{g_t}} \tag{4}$$

where $\sigma^2_{g_{jt}}$ is the genetic variance of the genetic factor $j$ contributing to trait $t$, and $\sigma^2_{g_t}$ the total genetic variance of trait $t$, based on

standardised factor loadings. Corresponding SEs were derived using the Delta method.

## Comparison of GRM-SEM with alternative methods

Besides GRM-SEM, multiple other approaches allow the modelling of the genomic covariance structure across phenotypes, including techniques such as genomic SEM[54]. For this study, we selected GRM-SEM for the following reasons: Genomic SEM, a linkage-disequilibrium-score-based technique[55], relies solely on genome-wide summary statistics. These statistics have to be powerful enough to identify genetic structure, requiring large effective sample sizes >10,000[56] that exceed those in SPARK and the SSC. In contrast, GRM-SEM, similar to GREML[17], uses a GRM derived from direct genotyping data to disentangle the full phenotypic covariance into a genetic and residual model part. The method requires similar sample sizes as for GREML (>2000)[57], matching those in this study. In addition, the fit of GRM-SEM models, but not genomic SEM models, can also be directly assessed against (i) a saturated model (with relative fit indices such as LRT, AIC and BIC) and (ii) the phenotypic covariance matrix (with absolute fit indices such as SRMR). Moreover, GRM-SEM allows for both genetic and residual covariance modelling, each with different structures, enhancing the model fit[14]. Lastly, the SE of the genomic covariance matrix can be directly inferred from the fitted GRM-SEM model, while SEs are approximated with jackknife procedures by genomic SEM, affecting the prediction accuracy of subsequent EFA analyses.

## Data-driven genomic covariance modelling approach

**Step (i): Describe full genetic architecture.** To describe the full genetic architecture, we fitted a saturated (Cholesky) model to the data in GRM-SEM.

**Step (ii): Predict the number of shared genetic factors.** Cholesky-derived genetic trait correlations provided input data to estimate $n_{AC}$, i.e. the number of genetic factors, using PCA via spectral decomposition[58] (*R:base*, v4.0.2). Eigenvalues of this genetic PCA were plotted as a scree plot and $n_{AC}$ was, eventually, estimated according to the Optimal Coordinate criterion (*R:nFactors*, v2.4.1)[59], applying a joint Kaiser's rule (eigenvalue >1)[60,61] and Cattell's scree test[62].

**Step (iii): Approximate genetic factor structure.** Given evidence for multiple genetic factors ($n_{AC} > 1$), we carried out EFA[63], predicting underlying genetic factor structures with *lavaan*[64] (*R:lavaan*, v0.6-10) software. As genetic trait covariance is not directly observable, we analysed the predicted genetic covariance matrix derived from a Cholesky model (step i). Factor solutions for this genetic EFA were estimated using a Diagonally Weighted Least Squares (DWLS) algorithm[65], i.e. a robust Weighted Least Squares (WLS) method that can be applied to skewed data where the likelihood function for any parameter θ is given as

$$l(\theta) = \frac{1}{2} tr[(S - \Sigma(\theta))W^{-1}] \tag{5}$$

where $S$ is the observed (here Cholesky predicted genetic covariance matrix) and $\Sigma$ the EFA model-implied genetic covariance matrix. Inverse weighting was carried out with a diagonal weight matrix W, based on the estimated variance $\tilde{V}$ of the genetic covariance $V_A$, as derived with a Cholesky model, where $W = \text{diag}(\tilde{V}(V_A))$. For comparison, we also carried out an unweighted least square estimation, where the identity matrix replaces W. Factors in *lavaan* were rotated using either (varimax) orthogonal or (oblimin) oblique rotation techniques. We opted for an EFA varimax model if the predicted genetic correlation between genetic factors by an EFA oblimin model was modest (i.e. $r \leq 0.32$[19] and thus ignorable) or if the EFA oblimin model produced a similar pattern of loadings as EFA

varimax[19] (Supplementary Data 8). In other words, when the EFA oblimin solution did not increase the simplicity of the model[19]. For sensitivity analyses, we also compared estimates of EFA *lavaan* with estimates of other EFA software (Supplementary Data 8) such as the *fa* function (*R:psych*, v2.2.3), which does not allow for inverse weighting[66].

**Step (iv): Define multi-factor models.** Using GRM-SEM, we fitted a multi-factor IPC model. Specifically, the factor loadings of the respective EFA model (step iii) were used to define starting values and constraints in the corresponding genetic part of the GRM-SEM. As a rule of thumb, GRM-SEM zero loadings (constraints) were defined based on EFA factor loadings of $|\lambda| < 0.10$[19,67]. Once fitted, we further trimmed the model by removing specific genetic GRM-SEM factor loadings near zero ($|\lambda| < 0.01$). The residual part of the model remained unchanged and was fitted as a Cholesky model. Note that an evaluation of EFA models based on model fit criteria established in observational research is not meaningful here, as the studied genetic covariance matrix (Cholesky) is estimated with an error that may result in negative uniqueness of the predicted genetic variance, violating modelling assumptions (known as a Heywood case)[68]. To confirm the independence of shared genetic factors, we fitted a bi-factor model. This model had a similar structure as the described multi-factor IPC model except that one factor was allowed to load on all phenotypes (see above).

**Step (v): Define one-factor models.** For sensitivity analysis, we fitted *a priori*-defined one-factor GRM-SEM IP and one-factor GRM-SEM IPC models (see above).

**Step (vi): Identify the best-fitting model.** We compared saturated (Cholesky), one-factor, bi-factor and multi-factor GRM-SEM models. The relative goodness-of-fit of each model, especially against the saturated model, was evaluated with LRT, AIC, and BIC fit indices and the absolute goodness-of-fit with SRMR indices (see above).

**Step (vii): Characterise identified shared genetic factors.** If computationally feasible, we added a mapping variable to characterise the factor structure of the best-fitting GRM-SEM model. In this study, the following mapping variables were available in SPARK.

- *Liability to Asperger.* We dichotomised ASD subcategory information in SPARK using Asperger as reference, as it was the ASD subcategory with the biggest sample size (see above). Individuals with Asperger diagnosis were coded as 1, individuals with childhood autism or PDD-NOS diagnosis were coded as 0, and individuals without an assigned subcategory were coded as missing. This variable was then transformed using deviance residuals. Note that low sample numbers and/or low $h^2_{SNP}$ of ASD liability prevented a more comprehensive modelling (Supplementary Fig. 19).
- *PGS$_{EA}$ mapping.* Consistent with current guidelines[69], we constructed PGS for EA within SPARK based on high-quality genome-wide imputed SNPs (Supplementary Methods 3), using available summary statistics from recent EA3 meta-GWAS[70]. For this purpose, we used PRS-CS software[71], which applies continuous-shrinkage parameters to adjust SNP effect sizes for linkage disequilibrium. Once SNP effect sizes were calculated in PRS-CS, PGS$_{EA}$ scores were calculated in PLINK[39] and, subsequently, Z-standardised.

### Phenotype SEM models
To compare genetic and phenotypic factor structures, we fitted a phenotypic SEM to the phenotypes included in the best-fitting GRM-SEM models. Adopting a data-driven modelling approach, analogous to our genetic pipeline (Fig. 1b), we first identified the number of phenotypic factors using eigenvalue decomposition of the phenotypic correlation matrix of the full sample. Using a split-half design (with two random

subsamples matched for sex and phenotype missingness), we subsequently conducted EFA on one half of the sample. Next, we retained factor loadings $|\lambda| > 0.10$ and confirmed the structure using CFA within the remaining half of the sample. We assessed the model fit based on the comparative fit index (CFI), the Tucker–Lewis index (TLI) and root mean square error of approximation (RMSEA) parameters[52]. Given good model fit[52] (i.e. CFI > 0.95, TLI > 0.95, RMSEA < 0.06), both halves of the cohort were combined again and a CFA model was fitted to the full sample.

### Simulation study
To evaluate the robustness of our data-driven genomic covariance modelling approach (see above, Fig. 1b), we carried out simulations. We assessed bias by comparing true values with GRM-SEM IPC factor loadings, as described in detail in the supplement (Supplementary Note 3, Supplementary Data 10–13). In brief, assuming multivariate normality, we simulated six-variate traits with either two shared genetic factors without correlation or two shared genetic factors with cross-loading as detailed by path models in Supplementary Fig. 14 and Supplementary Fig. 15, respectively, across 20 replicates. Each six-variate trait was based on Z-standardised phenotypes with 2000 individuals per phenotype and (for simplicity) 5000 causal *loci*, to increase power. Besides the median estimate, simulation performance measures included the median bias, the median empirical standard error (empSE) and coverage of 95%-confidence intervals (such that the estimated 95%-confidence interval contains the true value), and the respective Monte Carlo SEs (MCSE).

### Multiple testing
A correction for multiple testing of estimated GRM-SEM factor loadings of our analysis is not directly applicable. We jointly analyse multiple phenotypes using a multivariate approach to comprehensively represent all shared genetic factors across the studied ASD phenotypic spectrum. GREML estimates for $h^2_{SNP}$ and $r_g$ within Stage I are not individually interpreted, given the preliminary character of these analyses. However, if a multiple testing adjustment for individual measures reported during Stage I was considered, an experiment-wide threshold of $p < 0.0015$ (0.05/34 independent measures) would need to be applied, as estimated with Matrix Spectral Decomposition (matSpD)[72], based on phenotypic score correlations.

### Reporting summary
Further information on the research design is available in the Nature Portfolio Reporting Summary linked to this article.

## Data availability
Genotype and phenotype data from the SPARK and SSC cohorts are available upon application and approval from the Simons Foundation Autism Research Initiative (SFARI) (https://www.sfari.org/resource/autism-cohorts/). Approved researchers can obtain the SPARK and SSC population dataset described in this study by applying at https://base.sfari.org. Detailed reasons for controlled access and details of any restrictions imposed on data use via data use agreements have been outlined in the RESEARCHER DISTRIBUTION AGREEMENT of the Simons Collection (https://s3.amazonaws.com/sf-web-assets-prod/wp-content/uploads/sites/2/2021/06/15165956/SFARI_RDA.pdf) to ensure compliance with data-protection. The timeframe for response of the SFARI Collection to data requests is rapid (usually <2 months). GWAS summary statistics for educational attainment (EA3, Lee et al. 2018) were accessed through the Social Science Genetic Association Consortium (SSGAC, https://thessgac.com/papers/3).

## Code availability
This study used openly available software and codes, specifically PLINK (PLINK v1.9, https://www.cog-genomics.org/plink/1.9/), PRScs (https://github.com/getian107/PRScs), GCTA-GREML (GCTA v1.93, https://

cnsgenomics.com/). We used the following R packages: stats 4.0.2, base 4.0.2, nFactors 2.4.1, psych 2.2.3, lavaan 0.6-10, grmsem 1.1.2 (https://gitlab.gwdg.de/beate.stpourcain/grmsem). Scripts used in this study are available in GitLab (https://gitlab.gwdg.de/pghc/disentangling-asd-heterogeneity-using-grm-sem/nature-communications-2023).

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

## Acknowledgements

The authors are grateful to all of the families in SPARK, the SPARK clinical sites and SPARK staff. We are grateful to all of the families at the participating Simons Simplex Collection (SSC) sites, as well as the principal investigators (A. Beaudet, R. Bernier, J. Constantino, E. Cook, E. Fombonne, D. Geschwind, R. Goin-Kochel, E. Hanson, D. Grice, A. Klin, D. Ledbetter, C. Lord, C. Martin, D. Martin, R. Maxim, J. Miles, O. Ousley, K. Pelphrey, B. Peterson, J. Piggot, C. Saulnier, M. State, W. Stone, J. Sutcliffe, C. Walsh, Z. Warren, E. Wijsman). We appreciate obtaining access to phenotypic and genetic data on SFARI Base. This work was supported by a grant from the Simons Foundation Autism Research Initiative (SFARI ID: 514787, PI B.S.P.) covering the work of L.D.H., M.B. and M.M.J.D., in addition, to support from the Max Planck Society. F.S., E.V., S.E.F. and B.S.P. were fully supported by the Max Planck Society. C.Y.S. was supported by the UK Medical Research Council (MRC) Integrative Epidemiology Unit at the University of Bristol (MC_UU_00011/3). J.B. was supported by the EU-AIMS and AIMS-2-TRIALS programmes which receive support from Innovative Medicines Initiative Joint Undertaking Grant No. 115300 and 777394, the resources of which are composed of financial contributions from the European Union's FP7 and Horizon2020 Programmes, and from the European Federation of Pharmaceutical Industries and Associations (EFPIA) companies' in-kind contributions, and AUTISM SPEAKS, Autistica and SFARI; and by the Horizon2020 supported programme CANDY (Grant No. 847818).

## Author contributions

B.S.P. designed and supervised the research. L.D.H., M.M.J.D. and B.S.P. analysed the genetic and phenotypic data. L.D.H., M.T.B and B.S.P regularly discussed the progress of the research. M.T.B., F.S., E.V, C.Y.S. and B.S.P. provided methodological support. J.B. and D.R. helped with interpretation of findings from a clinical perspective. B.V. and A.K. helped with the interpretation of findings from a methodological perspective. L.D.H. and B.S.P. wrote the manuscript. L.D.H, M.T.B, F.S, M.M.J.D, E.V, C.Y.S., A.K., J.B., B.V., S.E.F., D.R. and B.S.P. read and commented on the manuscript.

## Funding

## Competing interests

The authors declare no competing interests.
