## [Peer Review File · Nature Communications]

Structural models of genome-wide covariance identify multiple common dimensions in autismREVIEWER COMMENTS

Reviewer #1 (Remarks to the Author):

de Hoyos and colleagues conduct a series of analyses to identify latent genetic dimensions in ASD. They first conduct the analyses using SPARK dataset and then using the SSC datasets. They identify three genetic latent factors in the SPARK dataset, which they partly replicate in the SSC dataset. I have a few comments that I hope the authors can address.

1. The use of GRM-SEM removes several interesting phenotypes due to the lack of significant SNP heritability. I wonder, if the authors can comment on why this approach is preferred over a more comprehensive SEM at a phenotypic level followed by genetic analyses using GRM, genetic correlation, and polygenic score analyses?
2. Linked, it is interesting that very limited number of the canonical diagnostic phenotypes for ASD are included in the final model (e.g., sameness behaviour). To what extent is the model truly representative of ASD vs representative of comorbidities including developmental delays?
3. Most of the genetic correlations were not statistically significant. Why did the authors only include GRMs that use only genotype data? Why did the authors not expand the analyses to also include the more recent release of SPARK datasets (they use data released in 2018)? Both will increase statistical power and may allow the inclusion of greater number of phenotypes.
4. I think the second factor represents motor issues rather than developmental issues. This is however, semantics.
5. Individuals with ID and or PTV de novo variants may have different factor structure. The SSC is enriched for this. How generalisable is the model in individuals with and without ID/de novo PTVs?
6. SCRL and SBEHAV are mentioned either in the text or in Figure 2B. I was not able to follow where SCRL is in the figure, and what SBEHAV refers to.
7. In general, the paper is difficult to follow for a general reader. I think quite a lot of it is because the methods are highly technical. And while I do not envy the authors in having to explain it, I think greater explanation of the terms and models directly in the results can be helpful.
8. Only 17 of the 47 phenotypes were included in the model. What were the phenotypes that were excluded? What are the consequences of these exclusions in how we can interpret the results? For instance, I do not find very many canonical ASD dimensions in the model.

Reviewer #2 (Remarks to the Author):

In this paper, de Hoyos and colleagues investigated the latent factor structure underlying genetic sharing of ASD symptoms in two separate cohorts. This is clearly an interesting topic and the authors did a good job backing up their conclusions with empirical evidence. But the paper will benefit from putting the discussion in the context of a broader literature and improving how some methodological details are reported and discussed. I have some specific comments.

1. Structural equation modeling has gained recent popularity in human genetics due to a series of papers which introduced Genomic SEM (or GSEM), a tool with an unfortunately similar name compared to the approach used in this paper. It would improve the paper if the authors could discuss the choice of using GRM-SEM instead of GSEM. To be clear, I think there are obvious arguments to make such as the expected imprecise h^2 and genetic correlation estimates based on GWAS summary data given low sample size, and the fact that individual-level data are in fact available in this study. Still, since GSEM has become somewhat mainstream in this type of studies, it is important to show the rationale of applying GRM-SEM.

2. Related to #1, it appears that GRM-SEM bases its inference on individual-level genotype and phenotype data, which inevitably involves modeling the non-genetic contribution on phenotypes. In contrast, GSEM completely ignores the non-genetic variance components. Given the fact that both genetic and non-genetic variance components are presenting in the model, a question then is what it means to have different latent factor structures underlying them for the same set of traits (e.g., shown in Fig 3B/E/H). The authors provided some previous work on this choice but it would be important to justify this setup and clarify whether this is for model convergence / computational convenience only and whether the results can still be interpreted given the setup. A related question is whether we would expect to see different results in GRM-SEM applications compared to just a SEM exercise based on phenotype data alone.

3. The application involving EA PGS is interesting but is also somewhat different from all other analyses done in this study, which is why some additional discussions on applying GRM-SEM to PGS data will be helpful. For example, what is the non-genetic factor underlying EA PGS (which I think is what E1 in Fig 5D denotes)? How predictive is this PGS and whether its imprecision will affect the GRM-SEM results? Also, why not using cognition data in SPARK for this analysis? Is it something not measured in SPARK?

4. In addition, the genetic correlation between EA/cognition and ASD is very relevant for the discussion about ASD phenotypic and genetic heterogeneity. There are many papers on this topic (e.g., PMID: 28504703 and 34493297). It would be helpful if the authors to clarify whether and how the results here contribute to our understanding of the positive EA-ASD genetic correlation (from GWAS) and the apparent negative cognition-ASD phenotypic correlation.

5. I initially thought the choice of "population representative" is a bizarre choice of phrase to describe this study because the authors only included samples with European ancestry in the analysis. Then I realized this was meant to contrast the simplex ascertained SSC cohort which the authors used for replication. Maybe consider rephrasing this?

6. LRT was used to compare different models. Are the models always nested in this type of comparison? If not, is LRT a justified statistical test? Related to this, some LRT p-values were 1 in this study which is bizarre because even if the null hypothesis were true or the sample size is very small, p-values are still expected to follow a uniform(0,1) distribution. Always having very large p-values seems to be a sign that the statistical tests were not done properly.

7. In section "Phenotype transformations", the authors stated "For categorical phenotypes and co-morbid disorders, we constructed deviance residuals as the difference between the logistic model fit and the fit of an ideal model". Clarify what an ideal model is here?

Response to reviewer comments

We thank the reviewers for this constructive feedback. Please find a detailed point-by-point response to each question raised by the reviewers below.

Reviewer #1 (Remarks to the Author)

de Hoyos and colleagues conduct a series of analyses to identify latent genetic dimensions in ASD. They first conduct the analyses using SPARK dataset and then using the SSC datasets. They identify three genetic latent factors in the SPARK dataset, which they partly replicate in the SSC dataset. I have a few comments that I hope the authors can address.

We thank the reviewer for their comments.

1.1. The use of GRM-SEM removes several interesting phenotypes due to the lack of significant SNP heritability. I wonder, if the authors can comment on why this approach is preferred over a more comprehensive SEM at a phenotypic level followed by genetic analyses using GRM, genetic correlation, and polygenic score analyses?

We thank the reviewer for the opportunity to address these questions. Previous research has focused on characterising ASD heterogeneity using SEM approaches at the phenotypic level (Warrier et al. 2022). In our work, we aim to understand ASD phenotypic heterogeneity, as explained by common genetic variation, studying samples of autistic individuals only. We have clarified the difference in strategies in the Introduction (page 3, paragraph 3):

“Thus, we model genomic covariance directly, in contrast to previous studies (8,9) that interrogate the genetic architecture in ASD-only samples through analyses of phenotypic factor structures.”

Although phenotypic relationships are fair representations of genetic relationships (Cheverud’s conjecture (Cheverud 1988)), they do not fully capture genetic links. For the comparison of phenotypic and genetic relationships, we have now fitted phenotypic SEM models to the traits included in the final GRM-SEM model (see also our detailed response to Reviewer 2, comment #2.2). Overall, when comparing phenotypic and genetic model structures, our GRM-SEM models identified genetic links and structures that were not reflected at the phenotypic level, highlighting the importance of modelling genomic covariance (Rebuttal Fig. 3 and 4 below). For example, in this study, we observe differences in phenotypic and genetic relationships with self-injurious behaviour. As such, modelling genomic covariance (as carried out in this work) instead of phenotypic covariance ensures the comprehensive representation of underlying common genetic structures.

We have now adjusted the manuscript as follows:

Results, page 6, paragraph 3:

“In line with Cheverud’s conjecture (21), which postulates that phenotypic relationships are likely to be fair estimates of their genetic counterparts, genetic dimensions largely matched corresponding phenotypic dimensions (Supplementary Fig. 6). Nonetheless, several differences between phenotypic and genetic structures became evident, such as for age of self-feeding and RBSR self-injurious behaviour. (...) These results leverage the importance of a data-driven genomic covariance modelling approach as genetic relationships may not be fully reflected by phenotypic relationships, given that the latter are also shaped by non-genetic/residual influences.”

Discussion, page 11, paragraph 3:

“Our study has multiple strengths, but also limitations. First, we developed a data-driven modelling approach that utilises directly genotyped genome-wide information and facilitates building accurate multi-dimensional models of genomic covariance without the need for summary statistics or the prediction of structure through phenotypic SEM. Although phenotypic structures were, overall, fair estimates of genetic structures (21), they could not fully capture underlying genetic relationships.”

Note that within GRM-SEM, we select measures based on h^2_{SNP} in order to identify phenotypes that are likely to have genetic contributions. Therefore, even if non-heritable phenotypes were included in our GRM-SEM approach (assuming that such model would converge), they would have little bearing on the identified genomic structures, given the lack of genetic contributions.

1.2. Linked, it is interesting that very limited number of the canonical diagnostic phenotypes for ASD are included in the final model (e.g. sameness behaviour). To what extent is the model truly representative of ASD vs representative of comorbidities including developmental delays?

Thank you for this interesting question. In our study, we investigate genetically predictable heterogeneity within ASD individuals only (case-only design). The “case-only” design has been previously adopted by several other groups (Warrier et al. 2022; Thomas et al. 2022). As we study ASD individuals only, our final model does not capture risk of ASD compared to a control group (i.e. case-control design). Only in comparison with a control population, we would capture “canonical” diagnostic ASD phenotypes, including difficulties in social communication and interactions as well as stereotyped, restricted and repetitive interests and behaviour (as defined in the DSM IV (American Psychiatric Association 1994)). Therefore, our study, adopting a case-only design, will not capture symptomatic differences between healthy controls and autistic individuals. Instead, our study examines and models the phenotypic differences across autistic individuals (phenotypic heterogeneity), and is, thus, representative of a spectrum of ASD diagnoses.

As discussed above, non-heritable phenotypes will have little bearing on the identified genomic structures, although lack of h^2_{SNP} may have additional interpretations within the context of a case-only study. As outlined within our amended manuscript (Results, page 4, paragraph 3):

“Notably, social and affective phenotypes showed little h^2_{SNP} (Supplementary Fig. 2), consistent with either phenotypic homogeneity (as social difficulties are present across the entire ASD spectrum), or little contributions of common genetic variation to phenotypic variation in an ASD case-only sample, or lack of power.”

1.3. Most of the genetic correlations were not statistically significant. Why did the authors only include GRMs that use only genotype data? Why did the authors not expand the analyses to also include the more recent release of SPARK datasets (they use data released in 2018)? Both will increase statistical power and may allow the inclusion of greater number of phenotypes.

(additional clarification, sent separately) They are now clarifying that they would like you to consider both imputed and genotyped data when generating GRMs (rather than just genotype data).

Thank you for the opportunity to clarify our modelling approach. To address the reviewer's concern, we have now carried out a comparison of our final GRM-SEM model based on GRMs derived from either direct genotype or imputed data, in both SPARK and SSC (see Rebuttal Fig. 1 and Rebuttal Fig. 2, respectively). These analyses demonstrate the consistency of findings for both models based on genotyped-data GRMs and imputed-data GRMs, as captured by overlapping SE and, thus, 95% CIs.

We have now clarified information on SPARK data within our work. Recent papers studying SPARK (Thomas et al. 2022; Antaki et al. 2022) have used the same 2019 phenotype data release (SPARK Collection Version 3) as investigated in this study. Regarding the genetic data, we used the SPARK 30K release (version 20181105), note that this release does not significantly differ from later releases used in recent papers (ver. 20181105 $N_{\text{ind}}=27,615$, of which 9,843 had ASD diagnosis; ver. 201909112 $N_{\text{ind}}=27,270$, of which 9,765 had an ASD diagnosis) (Warrier et al. 2022; Antaki et al. 2022). Given the similarity in SPARK data researched by us and other groups (based on recent publications) and the computationally expensive modelling approach applied in this work, we adhered to the data sets selected in this study, as we believe that the differences in underlying genomic structures will be small.

Rebuttal Fig. 1 Comparison of GRM-SEM estimates for the best-fitting model in the SPARK sample. Estimates for imputed-data GRMs and genotyped-data GRMs are shown in purple and black, respectively. Error bars indicate standard errors.

Rebuttal Fig. 2 Comparison of GRM-SEM estimates for the best-fitting model in the SSC sample. Estimates for imputed-data GRMs and genotyped-data GRMs are shown in purple and black, respectively. Error bars indicate standard errors.

1.4. I think the second factor represents motor issues rather than developmental issues. This is however, semantics.

We agree with the reviewer’s suggestion and amended the manuscript accordingly. Specifically, we have now renamed the factor to “developmental motor delay”.

1.5. Individuals with ID and or PTV *de novo* variants may have different factor structure. The SSC is enriched for this. How generalisable is the model in individuals with and without ID/*de novo* PTVs?

We thank the reviewer for raising this point. We have addressed this question in detail as part of our follow-up analyses in the SSC. The aim of this analysis was to study the comparability of structures identified in SPARK (Fig. 3) and the SSC (Fig. 4). The former is a community ASD sample, whereas the latter is a sample of simplex families that were recruited according to strict recruitment criteria. We outlined similarities and differences in model structure between SPARK and the SSC in the Discussion (page 9, paragraph 3):

“For SPARK, a community ASD sample, we identified three common genetic factors explaining predominantly variation in language performance, developmental motor delay and behavioural problems, respectively. Within the SSC, a sample of simplex-only families, we uncovered structural similarities, indicating conceptual replication. Hence, our findings not only strengthen the evidence for common genetic contributions to phenotypic variation in ASD (8,9,12), but also offer insight into the multi-dimensional genetic architecture. Specifically, we show that the majority of h^2_{SNP} in ASD-only samples can be explained by shared genetic factors, for most phenotypes investigated. The major difference across cohorts concerned the genetic relationship between language and behavioural phenotypes. While genetic factors of language performance and behaviour were unrelated in a community ASD sample (SPARK), the underlying phenotypes were strongly genetically linked in simplex ASD (SSC) and captured by a single dimension, suggesting ascertainment-specific association patterns.”

However, it is important to note that differences in identified genetic models within each cohort may not only reflect differences in common genetic architectures in individuals with and without ID/*de novo* PTVs, where the latter are enriched in the SSC but also differences in recruitment. As outlined in the Discussion (page 11, paragraph 2):

“Simplex ASD, compared to multiplex ASD, is more often related to *de novo* mutations (11,23). Our findings may, therefore, present aetiological differences unique to simplex ASD, consistent with qualitative differences in the common genetic architecture of ASD individuals carrying *de novo* variants (5,6). Alternatively, genetic links between behaviour and language phenotypes in the SSC might, to some degree, be a consequence of collider bias (33). Simplex families are recruited following strict ascertainment schemes (16). Collider bias can arise when two measures, such as behaviour and language/cognition, are independently related to a third variable, such as common genetic variation, and that third variable is conditioned upon (33). Here, the preferential ascertainment of simplex families depleted for inherited genetic risk (34), including common variation, may introduce artificial genetic relationships between behaviour and language/cognition.”

We, furthermore, attempted to stratify SPARK according to individuals with and without ID (with an increased likelihood of *de novo* mutations) in order to repeat GRM-SEM within different subgroups. Within SPARK, out of the 5,331 individuals, there are 1,256 individuals without ID and 964 individuals with ID, based on the definition of ID from a previous paper using SPARK data (Kuo et al. 2022). However, fitting structural models to individuals with and without ID within SPARK is currently not feasible. Given the low sample numbers for these subgroups, fitted models resulted in few detectable factor structures (data not shown). However, principal component analysis suggested that also within this subgroup there are likely to be three genetic dimensions.

We adjusted the Discussion (page 12, paragraph 1) as follows:

“Similarly, studying subgroups of individuals with and without ID was not yet feasible, given limited power (e.g. based on simple GREML models, the power for subgroup analyses with $N \sim 1200$ and h^2_{SNP} of 0.2, as observed in this study, is 0.11). Stratifying GRM-SEM models across common, rare and *de novo* carriers in sufficiently powered samples, accounting for differences between males and females, will shed further light onto the complex links between genetic and phenotypic heterogeneity in ASD, as part of future studies.”

However, irrespective of differences between SPARK and the SSC, we uncovered structural similarities across cohorts, indicating conceptual replication.

Discussion, page 9, paragraph 4:

“Across both cohorts, we found evidence for an independent language performance factor, as validated through association with higher liability to Asperger in SPARK. Although an Asperger ASD subcategory diagnosis is not included in the DSM-5 anymore, our findings confirm that autistic individuals differ considerably in their language presentation (27). While some children with ASD reach intact structural language skills, others are delayed or never master functional spoken language (27). Here, our analyses uncovered genomic covariance between (higher) language level and (earlier) age of self-feeding with a spoon, an important personal-social developmental milestone which typically developing children master at about 15-18 months (28,29).”

Discussion, page 10, paragraph 2:

“We also found robust evidence for a genetic factor that is related to developmental motor delay within SPARK and the SSC, explaining genetic variation underlying growth, such as the age of crawling, a developmental milestone children typically master between 9-18 months of age (31).”

1.6. SCRL and SBEHAV are mentioned either in the text or in Figure 2B. I was not able to follow where SCRL is in the figure, and what SBEHAV refers to.

We thank the reviewer for this comment. Within the previous version of our manuscript, we referred to different phenotypic subsets that were fitted as part of our modelling approach. To increase clarity, we have now re-structured the manuscript and moved information about intermediate model-fitting stages to the supplement, including subset GRM-SEM models (Supplementary Notes 1 and 2). The main manuscript represents now the final models for SPARK and the SSC only.

Given the reviewer’s comment, we would like to note, however, that the abbreviation SBEHAV has not been used in our manuscript. Regarding the abbreviation SCRL, this refers to the subset S_{CRL} , which contained phenotypes that were consistent with genetic correlations with age of crawling (Supplementary Fig. 3b).

1.7. In general, the paper is difficult to follow for a general reader. I think quite a lot of it is because the methods are highly technical. And while I do not envy the authors in having to explain it, I think greater explanation of the terms and models directly in the results can be helpful.

We agree with the comment of the reviewer. We have now fully re-structured the results section of the main manuscript and focussed on the final models in SPARK and the SSC only, moving a large portion of the details and intermediate models to either the methods section (“Study design” and “Data-driven genomic covariance modelling approach” in Methods) or the supplement (Supplementary Notes 1 and 2).

1.8. Only 17 of the 47 phenotypes were included in the model. What were the phenotypes that were excluded? What are the consequences of these exclusions in how we can interpret the results? For instance, I do not find very many canonical ASD dimensions in the model.

We apologise if this was not clear. We provide a full description of the phenotypes studied in SPARK in Supplementary Table 1. We also provide a short description of the domains that were studied in the Methods section (“Phenotypes” in Methods) and in the results section (see below):

Results, page 4, paragraph 3:

“... screening a wide range of language, cognitive, motor, developmental, affective, behavioural and social phenotypes (<https://www.sfari.org/spark-demographic-and-clinical-information>).”

Results, page 4, paragraph 3:

“From an initial set of 47 phenotypes (Methods, Supplementary Table 1, Supplementary Fig. 1), we retained 17 phenotypes representing five ASD domains: language/cognition, general behaviour, developmental

milestones, motor, and repetitive behavioural features (Fig. 2a). Notably, social and affective phenotypes showed little h^2_{SNP} (Supplementary Fig. 2) ...”

Methods, page 13, paragraph 3:

“We studied 47 parent-reported measures of ASD phenotypes and co-morbid disorders/disabilities spanning the domains of language and cognition (9 measures), general behaviour (9 measures), repetitive behaviour (7 measures), social (2 measures) and motor abilities (6 measures), as well as affective disorders (3 measures) and developmental milestones (11 measures). Phenotypes were extracted from the Basic Medical Screening Questionnaire (BMS), the Social Communication Questionnaire-Lifetime (SCQ) (36), the SPARK Background History Questionnaire (BGHX), the Repetitive Behaviours Scale-Revised (RBSR) (37), and the Developmental Coordination Disorder Questionnaire (DCDQ) (38) (Supplementary Fig. 1, Supplementary Methods 1).”

The exclusion of phenotypes due to low h^2_{SNP} has little bearing on the presented multi-variate genomic models, as phenotypic variation in these measures is not captured by genomic covariance. As outlined above (see Reviewer 1, comment #1.2), canonical ASD dimensions are symptoms which are defined with respect to a healthy control population. Here, we investigated, however, symptom variations among individuals with ASD only.

Reviewer #2 (Remarks to the Author):

In this paper, de Hoyos and colleagues investigated the latent factor structure underlying genetic sharing of ASD symptoms in two separate cohorts. This is clearly an interesting topic and the authors did a good job backing up their conclusions with empirical evidence. But the paper will benefit from putting the discussion in the context of a broader literature and improving how some methodological details are reported and discussed. I have some specific comments.

We thank the reviewer for their supportive comments.

2.1. Structural equation modelling has gained recent popularity in human genetics due to a series of papers which introduced Genomic SEM (or GSEM), a tool with an unfortunately similar name compared to the approach used in this paper. It would improve the paper if the authors could discuss the choice of using GRM-SEM instead of GSEM. To be clear, I think there are obvious arguments to make such as the expected imprecise h^2 and genetic correlation estimates based on GWAS summary data given low sample size, and the fact that individual-level data are in fact available in this study. Still, since GSEM has become somewhat mainstream in this type of studies, it is important to show the rationale of applying GRM-SEM.

We agree with the reviewer’s suggestion. We have now explained our reasoning to select GRM-SEM and not genomic SEM in this paper, and added these details to the Methods section of our manuscript (page 19, paragraph 1):

“Comparison of GRM-SEM with alternative methods

Besides GRM-SEM, multiple other approaches allow the modelling of the genomic covariance structure across phenotypes, including techniques such as genomic SEM (50). For this study, we selected GRM-SEM for the following reasons: Genomic SEM, a linkage-disequilibrium-score-based technique (51), relies solely on genome-wide summary statistics. These statistics have to be powerful enough to identify genetic structure, requiring large effective sample sizes $>10k$ (52) that exceed those in SPARK and the SSC. In contrast, GRM-SEM, similar to GREML (17), uses a GRM derived from direct genotyping data to disentangle the full phenotypic covariance into a genetic and residual model part. The method requires similar sample sizes as for GREML ($>2k$) (53), matching those in this study. In addition, the fit of GRM-SEM models, but not genomic SEM models, can also be directly assessed against (i) a saturated model (with relative fit indices such as LRT, AIC and BIC) and (ii) the phenotypic covariance matrix (with absolute fit indices such as SRMR). Moreover, GRM-SEM allows for both genetic and residual covariance modelling, each with different structures, enhancing the model fit (14). Lastly, the SE of the genomic covariance matrix can be directly inferred from the

fitted GRM-SEM model, while SEs are approximated with jackknife procedures by genomic SEM, affecting the prediction accuracy of subsequent EFA analyses. ”

Please note that our group introduced the term GSEM for genomic covariance modelling in 2018 (St Pourcain et al. 2018). We have since re-named the approach to GRM-SEM to avoid confusion with the genomic SEM approach.

2.2. Related to #1, it appears that GRM-SEM bases its inference on individual-level genotype and phenotype data, which inevitably involves modelling the non-genetic contribution on phenotypes. In contrast, GSEM completely ignores the non-genetic variance components. Given the fact that both genetic and non-genetic variance components are presenting in the model, a question then is what it means to have different latent factor structures underlying them for the same set of traits (e.g., shown in Fig 3B/E/H). The authors provided some previous work on this choice but it would be important to justify this setup and clarify whether this is for model convergence / computational convenience only and whether the results can still be interpreted given the setup. A related question is whether we would expect to see different results in GRM-SEM applications compared to just a SEM exercise based on phenotype data alone.

We thank the reviewer for this insightful question. GRM-SEM dissects multivariate phenotypic covariance similar to a GREML approach (Yang et al. 2011), where the residual variation represents the phenotypic variation that is unaccounted for by the genetic model. While most researchers focus on the common genetic aspects of the model, the residual component captures everything else (rare, non-additive or un-tagged genetic influences, environmental risk factors, and random error (Yang et al. 2011)) and is, thus, an intrinsic part of GRM-SEM. Therefore, residual variation is not fitted for convenience or model convergence. In fact, the model fit can only be accurately assessed once the residual part is taken into account, as both genetic and residual covariance contribute to phenotypic covariance. Specifically, within GRM-SEM, we guide model selection through comparison against a saturated model fitted to phenotypic data (i.e. using relative fit indices such as LRT, AIC and BIC). In addition, we can select models based on measures of absolute fit. For this, we have now implemented the Standardised Root Mean Square Residual (SRMR), which captures the similarity in GRM-SEM-predicted compared to observed phenotypic covariance. Neither approach is feasible in genomic SEM due to the lack of raw phenotype information, given the use of summary statistics and. Nevertheless, the genetic components of GRM-SEM and genomic SEM can be compared relative to each other.

Results (page 5, paragraph 1):

“Eventually, this information was used to build a hybrid Independent Pathway/Cholesky (IPC) GRM-SEM model, where the structure is only modelled within the genetic part of the data, while the residual part is always fitted to a saturated (Cholesky) model (Methods). IPC models have previously been shown to provide a superior model fit compared to other *a priori*-defined models (14)”

Methods (page 18, paragraph 2):

“The relative goodness-of-fit for each model was evaluated with likelihood ratio tests (LRTs), the Akaike information criterion (AIC) and the Bayesian information criterion (BIC) (47). The absolute goodness-of-fit was assessed with the standardised root mean square residual (SRMR)(48), as the standardised difference between the observed and predicted correlation, accounting for the degrees of freedom in GRM-SEM. SRMR values below a cut-off value of 0.08 indicate a good model fit (48).”

Results (page 8, paragraph 3):

“Second, we corroborated GRM-SEM predicted h^2_{SNP} (Supplementary Fig. 10) and rg (Supplementary Fig. 11) estimates for the best-fitting GRM-SEM IPC models through comparisons with corresponding GREML estimates that showed consistent 95% CIs.(...) Third, we illustrated the superiority in model fit for the best-fitting GRM-SEM IPC models by comparing their fit with exploratory GRM-SEM models (Supplementary Table 4), such as one-factor independent pathway and one-factor IPC models (Methods, Supplementary Fig. 12). ”

To address the reviewer’s last question, we have fitted an SEM model to the phenotype data alone. For SPARK, we have added the results of this analysis to Supplementary Fig. 6, also shown below (Rebuttal Fig. 3). For the SSC, we were not able to fit a phenotypic SEM model using a split-half approach, due to convergence

problems. For SPARK, when comparing genetic (Rebuttal Fig. 4, Fig. 3) with phenotypic (Rebuttal Fig. 3, Supplementary Fig. 6) structures, differences in the association between the age of self-feeding with a spoon and self-injurious behaviour (Results, page 6, paragraph 3) could be observed:

“In line with Cheverud’s conjecture (21), which postulates that phenotypic relationships are likely to be fair estimates of their genetic counterparts, genetic dimensions largely matched corresponding phenotypic dimensions (Supplementary Fig. 6). Nonetheless, several differences between phenotypic and genetic structures became evident, such as for age of self-feeding and RBSR self-injurious behaviour. For example, genetic variation in age of self-feeding was explained by the language performance factor (genetic model: $\lambda_{\text{lang}}=-0.38$, $SE=0.14$, Fig. 3), while phenotypic variation was accounted for by the developmental motor delay factor (phenotypic model: $\lambda_{\text{dev}}=0.50$, $SE=0.03$, Supplementary Fig. 6). Similarly, RBSR self-injurious behaviour was genetically linked to the developmental motor delay factor (genetic model: $\lambda_{\text{dev}}=0.36$, $SE=0.10$, Fig. 3), while sharing phenotypic variation with the behavioural problems factor (phenotypic model: $\lambda_{\text{beh}}=0.66$, $SE=0.04$, Supplementary Fig. 6). These results leverage the importance of a data-driven genomic covariance modelling approach as genetic relationships may not be fully reflected by phenotypic relationships, given that the latter are also shaped by non-genetic/residual influences.”

Rebuttal Fig. 3 Phenotypic SEM model in the SPARK sample.

Rebuttal Fig. 4 GRM-SEM model in the SPARK sample

2.3. The application involving EA PGS is interesting but is also somewhat different from all other analyses done in this study, which is why some additional discussions on applying GRM-SEM to PGS data will be helpful. For example, what is the non-genetic factor underlying EA PGS (which I think is what E1 in Fig 5D denotes)? How predictive is this PGS and whether its imprecision will affect the GRM-SEM results? Also, why not using cognition data in SPARK for this analysis? Is it something not measured in SPARK?

As outlined in our manuscript, we added information on Asperger and EA PGS as external reference points to guide the interpretation of identified genetic model structures within a study of individuals with ASD only. The need for an external reference point arises as we study an ASD-only sample, therefore an external reference helps interpret the factor structure compared to a general population sample. For example, individuals with a high cognitive load will differ in their comorbid symptom structure when studied within an ASD sample (where they correspond largely to individuals with Asperger, based on a DSM-IV diagnosis) compared to a general-population sample, containing mostly healthy individuals.

Results (page 8, paragraph 1):

“To enhance the interpretability of identified genetic structures in SPARK, we mapped further variables onto the genetic model structure (Methods). Specifically, we investigated the association between the identified factors and (i) liability to Asperger (compared to other ASD subcategories) (Fig. 5a-c) and (ii) PGS for educational attainment (PGS_{EA}) (Fig. 5d-f). ASD subcategory information (DSM-IV-based) can provide a clinical reference to account for different phenotypic presentations in ASD. Here, it can guide the interpretation of identified genetic dimensions, as genetic liability to Asperger presents a form of autism without significant impairments in language and cognitive development (24). PGS_{EA} presents a genetic correlate of cognitive functioning (25), but also socio-economic status, including non-cognitive factors such as health and longevity (26).”

To answer the reviewer’s first question, here, we dissect variation of PGS_{EA} into genetic (shared and specific) and residual (non-genetic) variation. The residual variance (E1) for PGS_{EA} is very small ($E1=0.61\%(0.49\%)$) compared to the genetic variance ($AS=96.57\%(1.91\%)$, $A_{lang}=0.24\%(0.54\%)$, $A_{dev}=0.02\%(0.14\%)$, $A_{beh}=2.53\%(1.98\%)$). This suggests that some PGS_{EA} variation is unrelated to genetic variance as captured by GRMs in ASD samples. For example, it is possible that a small part of PGS_{EA} variance represents noise, or that GRMs in ASD samples capture marginally different common variation compared to individuals included in EA meta-GWAS discovery analyses (Lee et al. 2018).

Regarding the reviewer’s second question, mapping PGS onto genetic models, including EA PGS, provides information about the structure of PGS that is not detectable with regression-based approaches. Within this study, we asked the question of whether identified shared genetic factors in SPARK can explain variation in PGS_{EA} . Therefore, we cannot assess the predictive ability of PGS. However, we may assume the reciprocity of associations given that variation in PGS_{EA} is near-fully explained by genetic variation. Therefore, an estimate of 2.79% explained variation in PGS_{EA} through shared factors (by A_{lang} , A_{dev} and A_{beh}) is consistent with typically observed explained variation in trait variance, based on cohorts with similar sample size (Selzam et al. 2017).

The reviewer is correct that information on cognitive performance is available in SPARK. As part of this study, we have investigated: intellectual disability, cognitive age level and cognitive impairment (Supplementary Fig. 2). Among these cognition-related phenotypes, only cognitive age level passed the h^2_{SNP} selection threshold of $p<0.1$. However, cognitive age level was solely related to the language performance factor (Supplementary Fig. 5b,e, Supplementary Note 2) and highly correlated with the remaining measures (cognitive age level and language level: GREML $r_g=0.87, SE=0.29$; cognitive age level and language disorder: GREML $r_g=-0.92, SE=0.53$; Supplementary Fig. 3), showing similar association patterns in structural models. To reduce the computational burden and collinearity, cognitive age level was therefore proxied by other measures and not included in the modelling process nor the final model. We attempted to set up subgroup models for individuals with intellectual disability (see Reviewer 1, comment #1.5). However, this was not feasible, as the power of such models was low resulting in little detectable genetic structure.

2.4. In addition, the genetic correlation between EA/cognition and ASD is very relevant for the discussion about ASD phenotypic and genetic heterogeneity. There are many papers on this topic (e.g., PMID: 28504703 and 34493297). It would be helpful if the authors clarify whether and how the results here contribute to our

understanding of the positive EA-ASD genetic correlation (from GWAS) and the apparent negative cognition-ASD phenotypic correlation.

We thank the reviewer for highlighting this important aspect of our manuscript. However, we are limited in placing our findings within the context of research conducted by Weiner and colleagues (PMID: 28504703 (Weiner et al. 2017)) and local genetic correlation analyses studying risk of ASD as carried out by Zhang and colleagues (PMID: 34493297 (Zhang et al. 2021)). Weiner and colleagues showed that common variant risk appears similarly relevant to ASD individuals with high and low IQ, and those with and without a strongly acting *de novo* mutation (Weiner et al. 2017). We, therefore, attempted to fit subgroup models for individuals with ID in SPARK. These models are, however, not feasible yet, given low power (see Reviewer 1, comment #1.5). Regarding the second paper, we are limited by differences in research design. While Zhang and colleagues (Zhang et al. 2021) study links between EA and risk of ASD (case-control design), this study adopts a case-only design and, therefore, we cannot draw any conclusions about risk of ASD. However, we have integrated our findings within the context of research carried out by Warrier and colleagues (Warrier et al. 2022) and Antaki and colleagues (Antaki et al. 2022), who also adopted a case-only design.

As outlined in the Results of our manuscript:

Results (page 8, paragraph 2):

“In contrast, PGS_{EA} were associated with reduced behavioural problems (Fig. 5d, $\lambda_{beh}=-0.16$, $SE=0.06$), conditional on the language performance and developmental motor delay dimensions. Consistent with previous research (8,9), genetic correlations of PGS_{EA} with behavioural measures such as sameness behaviour were inverse (Fig. 5f, $GRM-SEM r_g=-0.16$, $SE=0.06$), strengthening support for links with repetitive behaviour.”

Within the Discussion, we provided further explanation of this finding (page 10, paragraph 3):

“As in previous research adopting a case-only design (8,9), the behavioural problems factor was inversely associated with PGS_{EA} in SPARK. Our analyses demonstrated that this association can be observed conditional on genetic links with the language performance or the developmental motor delay factor, neither of which were related to PGS_{EA} . Thus, our findings suggest that behavioural problems within a community ASD sample vary primarily with non-cognitive correlates of socio-economic status.”

2.5. I initially thought the choice of "population representative" is a bizarre choice of phrase to describe this study because the authors only included samples with European ancestry in the analysis. Then I realized this was meant to contrast the simplex ascertained SSC cohort which the authors used for replication. Maybe consider rephrasing this?

We agree with the reviewer's comment. We have now rephrased the "population representative" with "autism community sample" to reflect the spectrum-wide recruitment of ASD patients within the SPARK sample, contrasting the simplex-ascertained SSC cohort.

Results (page 4, paragraph 2):

“... ASD community samples, i.e. unselected ASD samples with a wide demographic, phenotypic and clinical spectrum.”

2.6. LRT was used to compare different models. Are the models always nested in this type of comparison? If not, is LRT a justified statistical test? Related to this, some LRT p-values were 1 in this study which is bizarre because even if the null hypothesis were true or the sample size is very small, p-values are still expected to follow a uniform(0,1) distribution. Always having very large p-values seems to be a sign that the statistical tests were not done properly.

The reviewer is correct that fitted models are nested. It has previously been shown that the independent pathway model is nested within the saturated Cholesky model, and, thus, a LRT is justified (Neale and Maes 2004). Using a data-driven GRM-SEM approach, we provide GRM-SEM with starting values and constraints that guide model

identification. Thus, the model fit of the identified model is close to the saturated model (see Rebuttal Table 1 and Table 1). As such p -values are not expected to follow a uniform(0,1) distribution. Note that LRT p -values are rounded to one given very small differences between the saturated model and the best-fitting model. We have now replaced “LRT $p=1$ ” with “LRT $p>0.99$ ”). Within Rebuttal Table 1 (Supplementary Table 7), it also can be seen that for other, *a priori*-defined models, especially one-factor IP models, the observed model fit is statistically different (i.e. worse) compared to the saturated (Cholesky) model. In addition to LRTs, AIC and BIC (i.e. relative measures of fit), we have now implemented also absolute measures of fit, i.e. the standardised root mean square residual (SRMR), demonstrating the excellent fit of identified models (Table 1). In addition, we carried out comparisons with GREML estimates. We adjusted the manuscript as outlined in our reply to Reviewer 2, comment #2.2.

LRT p -values near one capture the similarity in fit between a saturated model and our best-fitting models, highlighting the strength of our data-driven genomic covariance modelling approach.

Rebuttal Table 1. Model fit comparison for final models in SPARK and SSC. Abbreviations: AIC (Akaike information criterion); BIC (Bayesian information criterion); IPC (Hybrid Independent Pathway (genetic part) / Cholesky (residual part) model); LRT (Likelihood ratio test); N_{par} (number of parameters), SRMR (standardised root mean square residual).

Model	Type	log-likelihood	N_{par}	AIC	BIC	SRMR	LRT _{Cholesky}	
							$\Delta\chi^2(\Delta\text{df})$	p
SPARK, $N_{\text{traits}}=8, N_{\text{ind}}=5279$								
Cholesky	saturated	-15248.61	72	30641.23	31114.37	0.002	-	
IP	one-factor	-15422.47	32	30908.94	31119.23	0.029	347.71 (40)	$<10^{-10}$
IPC	one-factor	-15262.09	52	30628.18	30969.90	0.002	26.96 (20)	0.14
Bi-factor	three-factor	-15249.97	62	30623.94	31031.37	0.002	2.71(10)	0.99
IPC best-fit	three-factor	-15250.96	53	30607.92	30956.21	0.002	4.69(19)	>0.99
SSC, $N_{\text{traits}}=8, N_{\text{ind}}=1940$								
Cholesky	saturated	-6342.50	72	12828.99	13230.07	0.008	-	
IP	one-factor	-6731.75	32	13527.49	13705.75	0.067	778.50 (40)	$<10^{-10}$
IPC	one-factor	-6349.53	52	12803.05	13092.72	0.008	14.06 (20)	0.83
Bi-factor	three-factor	-6342.59	63	12811.18	13162.12	0.014	0.19(9)	>0.99
IPC best-fit	three-factor	-6342.60	53	12791.19	13086.43	0.017	0.20(19)	>0.99

2.7. In section "Phenotype transformations", the authors stated "For categorical phenotypes and co-morbid disorders, we constructed deviance residuals as the difference between the logistic model fit and the fit of an ideal model". Clarify what an ideal model is here?

We apologise for this oversight. We have now rephrased “ideal model” into the statically accurate term “saturated” model. We have also adapted the “phenotype transformations” section to make clear how we derived these scores (page 14, paragraph 4):

“Categorical variables. After adjusting for covariates, deviance residuals were constructed by extracting the logistic model residuals using the *resid()* function (*R:stats* v4.0.2). Deviance residuals are computed as the difference between the logistic model fit to the data against the fit of a saturated model.”

REVIEWER COMMENTS

Reviewer #1 (Remarks to the Author):

I thank the authors for a comprehensive revision of the manuscript. I'd particularly like to commend the authors on making this paper a lot easier to read and more accessible to readers. The authors have conducted several additional analyses to bolster their findings. Although I'm largely satisfied with the revisions, I still have a few additional comments for the authors to consider.

1. With regard to my previous point about comparing phenotypic vs GRM-SEM models, it looks like the authors have conducted SEM models using the final list of phenotypes in the GRM-SEM model. Whilst this is useful, it does not help us to find out if using SEM on all phenotypes (not just the final modelled ones), followed by genetic analyses of the latent structures provides better interpretation of the common dimensions in autism. I request the authors to conduct this additional analyses to contextualise the utility of GRM-SEM compared to traditional SEM methods followed by genetic analyses.

2. The findings with EA are interesting, but potentially complicated by the fact that the vast majority of EA GWAS effects act indirectly (Okbay et al., Nature Genetics, 2022). As such, it is really unclear how to interpret these findings - are the associations due to the direct effects of the child's genotype, an effect of parental genotypes or a combination of both? It will be helpful if the authors could acknowledge this in the discussion.

3. The authors have largely used "ASD", although in certain instances, they have used "autism". Language is a highly contentious issue within the autism communities. Whilst I don't want to make any recommendations on which terms to use, I would like the authors to think a bit more carefully about their choice of terms and perhaps a few lines to justify their choice of term to the readers. Please note, this is not just with reference to the term "ASD" or "autism" but the wider use of terms like risk.

Response to reviewer comments

We thank the reviewer for their positive comments. Please find a detailed point-by-point response to each question raised by the reviewer below.

Reviewer #1 (Remarks to the Author)

I thank the authors for a comprehensive revision of the manuscript. I'd particularly like to commend the authors on making this paper a lot easier to read and more accessible to readers. The authors have conducted several additional analyses to bolster their findings. Although I'm largely satisfied with the revisions, I still have a few additional comments for the authors to consider.

1. With regard to my previous point about comparing phenotypic vs GRM-SEM models, it looks like the authors have conducted SEM models using the final list of phenotypes in the GRM-SEM model. Whilst this is useful, it does not help us to find out if using SEM on all phenotypes (not just the final modelled ones), followed by genetic analyses of the latent structures provides better interpretation of the common dimensions in autism. I request the authors to conduct this additional analyses to contextualise the utility of GRM-SEM compared to traditional SEM methods followed by genetic analyses.

We thank the reviewer for the suggestion.

We respectfully disagree with the reviewer that the aim of our paper is *to find out if using SEM on all phenotypes (not just the final modelled ones), followed by genetic analyses of the latent structures provides better interpretation of the common dimensions in autism*. The aim of our paper is to dissect the h^2_{SNP} of ASD phenotypes, in full, in order to understand the underlying multivariate genetic architecture as a structure of shared and specific genomic variance contributions. To exemplify this, we have visualised Fig. 3c as the proportion of h^2_{SNP} of each phenotype that is explained by the common genetic factors for the SPARK model (Rebuttal Fig. 1).

Rebuttal Fig. 1 Genetic variance plot of the SPARK model (Fig 3). The y-axis represents the proportion of each phenotype's h^2_{SNP} that is explained by the three shared genetic factors (A_{lang} , A_{dev} , A_{beh}) or their own specific genetic factor (AS).

We have clarified our aim in the introduction (page 3, paragraph 3):

“In this study, we aim to understand whether phenotypic heterogeneity in ASD can be explained by heterogeneity in common genetic effects by studying autistic individuals from large ASD cohorts. To do so, we fully dissect the h^2_{SNP} of ASD phenotypes into shared and specific genomic variance contributions, as implemented in genetic-relationship-matrix (GRM) structural equation modelling (GRM-SEM) (13,14). [...] Therefore, GRM-SEM allows the direct modelling of the genomic covariance, in contrast to previous studies (8,9) that interrogate the genetic architecture in ASD through analyses of phenotypic factor structures followed by genetic association analyses.”

Rebuttal Fig. 2 Differences between (a) phenotypic SEM and (b) GRM-SEM approach. **a**, Phenotypic SEM approach. When extracting phenotypic factor scores for subsequent genetic analyses, we can only extract the information of the shared phenotypic variance component (✓), while genetic information in the specific variance components is lost (✗). **b**, GRM-SEM approach, including a genetic and a residual model. Note that the residual model has been simplified for clarity. In this study, the residual GRM-SEM model is implemented as a Cholesky decomposition.

Our aim is not compatible with a SEM approach at the phenotypic level due to underlying differences in the nature of the identified factors (genetic vs phenotypic). We highlighted inherent differences in shared variation extracted by phenotypic SEM and GRM-SEM in Rebuttal Fig. 2. A phenotypic SEM approach followed by genetic association analysis will focus on the analysis of phenotypic factor scores capturing shared genetic and residual variation only (green tickmark, ✓), while specific phenotypic variation, including genetic variation is lost (red cross, ✗). Thus, a phenotypic SEM approach followed by subsequent genetic analysis of factor scores does not allow dissecting the full h^2_{SNP} of ASD phenotypes as specific genetic factors are lost. In addition, the contribution of shared genetic factors to shared phenotypic factors is not uniform and will vary across different phenotypes. In contrast, GRM-SEM allows dissecting the full h^2_{SNP} into shared and specific genetic factors (Rebuttal Fig. 1). More specifically, it estimates the contribution of shared genetic variance across multiple genetic dimensions in addition to specific genetic variance.

We agree with the reviewer that finding an interpretation of the latent dimensions in autism across all available phenotypes is an interesting research question. However, it has been already implemented in two previous studies of the SPARK sample (Warrier et al.¹ and Taylor et al.²)

To address the reviewer's request and to highlight differences between a genetic and a phenotypic SEM approach, we performed a phenotypic factor analysis across the initial set of 47 phenotypes in SPARK. Note that we removed DCDQ and RBSR total scores to avoid collinearity with their respective subscales.

Using the pipeline described in our manuscript ("Phenotype SEM models" section, Methods, page 21) we identified a 12-factor model with correlated factors and an acceptable model fit (oblimin rotation, CFI = 0.93, TLI = 0.92, RMSEA = 0.031, SRMR = 0.068). We extracted factor scores (using *lavPredict* function in *lavaan*) using information from individuals with less than 50% missingness. Subsequently, we carried out a GREML analysis as described in "Univariate and bivariate genetic variance analyses" (Methods, page 16). Four out of the 12 factors were heritable ($h^2_{\text{SNP}} p < 0.05$). Heritable phenotypic factors (Rebuttal Fig. 3) described variation across repetitive behaviours from the RBSR questionnaire

Rebuttal Fig. 3 Phenotypic SEM analysis followed by genetic analysis across the 47 SPARK phenotypes included in this study. **a** GREML analysis of heritable factor scores. Heritability is shown in the diagonal and bivariate genetic correlations in the off diagonal. **b** Confirmatory factor analysis (CFA) factor loadings for heritable factors. A black square indicates factor loadings $|\lambda| > 0.3$. **a,b** Values for GREML and CFA analysis are shown with their standard error in parenthesis.

(F3), motor measures from the DCDQ questionnaire (F4), motor developmental milestones from the BGHX questionnaire (F5) and language and cognition measures (F11).

We observed similarities (Rebuttal Fig. 3, Rebuttal Fig. 4) between the heritable phenotypic factors and the identified GRM-SEM genetic factors. Phenotypic factor scores and GRM-SEM factors were similar for:

- A language factor: The phenotypic factor underlying language and cognition measures (F11) aligns with the language performance genetic factor (A_{lang}).
- Motor and developmental milestones: The phenotypic factors underlying motor (F4) and motor developmental milestones (F5) align with the developmental motor delay genetic factor (A_{dev}). Notably, these two phenotypic factors are genetically identical (GREML $rg_{F4,F5} = -1.00$, $SE = 0.31$).
- A repetitive behaviour factor. The phenotypic factor explaining variation across repetitive behaviours (F3) aligns with the behavioural problems genetic factor (A_{beh}).

Rebuttal Fig. 4 Phenotypic Pearson correlations between heritable phenotypic factor scores (across the 47 SPARK phenotypes included in this study, Rebuttal Fig. 3) and phenotypic factor scores from final GRM-SEM model (P_{lang} , P_{dev} and P_{beh} , Supplementary Fig. 6).

Differences between phenotypic SEM and GRM-SEM analysis may arise due to several reasons:

- The variance modelled in both approaches differs. Phenotypic factors are identified to capture using shared phenotypic variation, including shared genetic and residual variation, while genetic factors capture shared genetic variation only.
- Related to (i), as phenotypic factors capture shared genetic + residual variance, they will only capture a fraction of shared genetic variance (which can be measured as a factor score's h^2_{SNP}). In contrast, in GRM-SEM genetic factors (specific + shared) explain the entirety of a phenotype's h^2_{SNP} .

- (iii) In GRM-SEM, we can quantify the contributions of each shared genetic factor to both the total genetic variance (h^2_{SNP}) and (ii) the total phenotypic variance of each investigated phenotype. In contrast, in phenotypic SEM (followed by genetic analysis), the genetic contributions of each factor cannot be traced back to the phenotypic variation anymore. Therefore, genetic links between phenotypic factor scores cannot be modelled.
- (iv) Genetic variance underlying phenotypes with different residual and genetic components may not be uniformly captured in phenotypic SEM³. In contrast, GRM-SEM separates these two (i.e. residual and genetic) components and allows for the identification of latent genetic structures using direct genotyping information, capturing the full genetic variance of all modelled phenotypes.
- (v) Overall, the aims of the analyses are substantially different. Whereas a SEM at the phenotypic level aims to identify the shared sources of phenotypic (residual + genetic) variation, a SEM at the genetic level aims to investigate the proportion of h^2_{SNP} of each trait that is shared across phenotypes.

For these reasons, a direct comparison of phenotypic SEM factor structures followed by genetic analyses and GRM-SEM is not feasible. Therefore, we have not included these analyses into our manuscript.

Irrespective of these inherent methodological differences, both phenotypic SEM (followed by genetic analysis) and GRM-SEM indicate that there are several latent dimensions in autism, including dimensions related to repetitive behaviours, motor development and language, which have a substantial genetic component as explained by common genetic variants.

2. The findings with EA are interesting, but potentially complicated by the fact that the vast majority of EA GWAS effects act indirectly (Okbay et al., Nature Genetics, 2022). As such, it is really unclear how to interpret these findings - are the associations due to the direct effects of the child's genotype, an effect of parental genotypes or a combination of both? It will be helpful if the authors could acknowledge this in the discussion.

We agree. We have amended the manuscript accordingly.

Discussion, page 11, paragraph 1:

“However, it is important to highlight that a large proportion of the PGS_{EA} genetic effects are not due to direct effects, but indirect effects (e.g. non-transmitted parental genetic influences), other forms of gene-environment correlation or assortative mating (33). Therefore, the nature and causality of PGS_{EA} associations cannot be determined from our analysis.”

3. The authors have largely used "ASD", although in certain instances, they have used "autism". Language is a highly contentious issue within the autism communities. Whilst I don't want to make any recommendations on which terms to use, I would like the authors to think a bit more carefully about their choice of terms and perhaps a few lines to justify their choice of term to the readers. Please note, this is not just with reference to the term "ASD" or "autism" but the wider use of terms like risk.

We agree with the reviewer. To explain our language choice to the readers, we have now added the following paragraph to the discussion.

Discussion, page 12, paragraph 3:

“Language choice

We are aware that the choice of language plays an important role in the autism community (36,37). While some individuals prefer person-first language (i.e. individuals with autism), others prefer identity-first language (i.e. autistic individuals). Based on the preferences of the SPARK community (38), we have used person-first (“individuals with autism” “individuals with ASD”) and identity-first (“autistic individuals” or “ASD individuals”) terms interchangeably. We acknowledge and respect each individual’s preference to identify themselves.”

Additionally, we have replaced terms like “risk”, “symptoms” and “affected” throughout the manuscript to avoid language with negative connotations. There is one instance left of the word “risk” in the introduction, where we refer here to “common risk alleles” for disorders (not ASD itself).

REFERENCES

1. Warrier, V. *et al.* Genetic correlates of phenotypic heterogeneity in autism. *Nat Genet* **54**, 1293–1304 (2022).
2. Thomas, T. R. *et al.* Clinical autism subscales have common genetic liabilities that are heritable, pleiotropic, and generalizable to the general population. *Transl Psychiatry* **12**, 1–14 (2022).
3. Williams, C. M. *et al.* Characterizing the phenotypic and genetic structure of psychopathology in UK Biobank. *medRxiv* (2023) doi:10.1101/2023.09.05.23295086.

REVIEWERS' COMMENTS

Reviewer #1 (Remarks to the Author):

I thank the authors for comprehensively addressing my comments. This is a really nice addition to the literature.